# Reduction of specific enterocytes from loss of intestinal LGR4 improves lipid metabolism in mice

Yuan Liang [1,4], Chao Luo [1,4], Lijun Sun[1], Tiange Feng [1], Wenzhen Yin[1], Yunhua Zhang[1], Michael W. Mulholland[2], Weizhen Zhang [1,2] ✉ & Yue Yin [3] ✉

Whether intestinal Leucine-rich repeat containing G-protein-coupled receptor 4 (LGR4) impacts nutrition absorption and energy homeostasis remains unknown. Here, we report that deficiency of *Lgr4* (*Lgr4$^{iKO}$*) in intestinal epithelium decreased the proportion of enterocytes selective for long-chain fatty acid absorption, leading to reduction in lipid absorption and subsequent improvement in lipid and glucose metabolism. Single-cell RNA sequencing demonstrates the heterogeneity of absorptive enterocytes, with a decrease in enterocytes selective for long-chain fatty acid-absorption and an increase in enterocytes selective for carbohydrate absorption in *Lgr4$^{iKO}$* mice. Activation of Notch signaling and concurrent inhibition of Wnt signaling are observed in the transgenes. Associated with these alterations is the substantial reduction in lipid absorption. Decrement in lipid absorption renders *Lgr4$^{iKO}$* mice resistant to high fat diet-induced obesity relevant to wild type littermates. Our study thus suggests that targeting intestinal LGR4 is a potential strategy for the intervention of obesity and liver steatosis.

Leucine-rich repeat containing G-protein-coupled receptor 4 (LGR4) is characterized by a long extracellular domain (ectodomain) that contains multiple copies of leucine-rich repeats[1]. It is widely expressed in tissues ranging from the digestive organs, embryo, hypothalamus, cartilage to hair follicles[2]. In the intestine, LGR4 is abundantly expressed and contributes to the maintenance of intestinal homeostasis. Global deficiency of *Lgr4* impairs the development of intestine characterized by absence of *Lgr5$^+$/Olfm4$^+$* intestinal stem cells at E16.5, inhibition of Wnt signaling, and blockade of intestinal epithelial cell proliferation and differentiation[3]. Interestingly, *Lgr4* deficiency appears to block the terminal differentiation of Paneth cells only[4]. Significant reduction in Paneth cells and subsequent aggravation of mucosal injury render the *Lgr*4 deficient mice susceptible to dextran sodium sulfate induced inflammatory bowel disease[4]. Whether intestinal LGR4 impacts nutrition absorption and energy homeostasis remains unknown.

Absorption of nutrients including fatty acid, amino acid and glucose by enterocytes substantially influences the global energy homeostasis[5]. Inhibition of lipid breakdown and absorption in intestine is efficient for the intervention of obesity and relevant metabolic disorders[6]. Traditionally, each absorptive enterocyte is considered being capable of absorbing multiple nutrients simultaneously. Our present study provides evidence challenging this classical concept, supporting the cellular heterogeneity of absorptive enterocytes. Using the single cell RNA sequencing analysis, we revealed that absorptive enterocytes contained three distinct populations: cells selective for long-chain fatty acid absorption, for carbohydrate absorption, and non-selective for long-chain fatty acid and carbohydrate absorption. Deficiency of *Lgr4* specifically in intestinal epithelium (*Lgr4$^{iKO}$*) significantly reduced the proportion of cells selective for long-chain fatty acid absorption. The reduction in enterocytes selective for long-chain

[1]Department of Physiology and Pathophysiology, School of Basic Medical Sciences, and State Key Laboratory of Vascular Homeostasis and Remodeling, Peking University, 100191 Beijing, China. [2]Department of Surgery, University of Michigan Medical Center, Ann Arbor, MI 48109-0346, USA. [3]Department of Pharmacology, School of Basic Medical Sciences, and State Key Laboratory of Vascular Homeostasis and Remodeling, Peking University, 100191 Beijing, China. [4]These authors contributed equally: Yuan Liang, Chao Luo. ✉e-mail: weizhenzhang@bjmu.edu.cn; yueyin@bjmu.edu.cn

fatty acid absorption substantially decreased lipid absorption, leading to subsequent improvement in global lipid and glucose metabolism.

## Results

### Deficiency of intestinal *Lgr4* reduces body weight and protects mice from HFD-induced obesity

*Lgr4* mRNA was detected in a wide variety of tissues, with high abundance in hypothalamus and digestive organs such as pancreas, stomach, jejunum, ileum, and liver (Supplementary Fig. 1a). To determine whether intestinal LGR4 regulates energy metabolism, we generated *Lgr4^iKO* mice within which *Lgr4* was specifically knocked out in intestinal epithelium (Supplementary Fig. 1b–d). Six-week-old male *Lgr4^iKO* mice and wild type (WT) littermates were fed normal chow diet (NCD) or 60% high fat diet (HFD) for 12 weeks. Compared with WT littermates, significant weight loss and smaller body size were observed in *Lgr4^iKO* mice fed either NCD or HFD (Fig. 1a, b). However, there was no significant difference in the body condition score (BCS) of *Lgr4^iKO* mice compared with WT littermates (Fig. 1c). Food intake was substantially reduced only in mice fed NCD but not in animals fed HFD, indicating that the weight loss is not entirely dependent on food intake (Supplementary Fig. 2a). Consistently, *Lgr4^iKO* mice displayed significantly less fat mass and lean mass (Fig. 1d, e). Fat weight and adipocyte size of subcutaneous white adipose tissue (sWAT) were significantly decreased. mRNA levels of beigeing marker gene *Ucp1* was strikingly increased in sWAT of *Lgr4^iKO* mice fed HFD (Fig. 1f, g). Similar findings were observed in the epididimal white adipose tissue (eWAT) (Fig. 1h, i). These results suggest that deficiency of intestinal *Lgr4* reduces body weight and protects mice from HFD-induced obesity.

To further elicit the reason of weight loss in *Lgr4^iKO* mice, cold exposure and metabolic cage experiments were performed. As shown in Supplementary Fig. 2b, c, rectal body temperature during the 4 °C cold exposure, physical activity, and respiratory quotient ($RQ = VCO_2/VO_2$) were not significantly altered by deficiency of intestinal *Lgr4*. These results indicate that weight loss of *Lgr4^iKO* mice was unlikely due to the alteration in thermogenesis and energy expenditure.

### Deficiency of intestinal *Lgr4* protects mice from HFD-induced liver steatosis

Next, we examined hepatic lipid metabolism in *Lgr4^iKO* mice. Liver weight and plasma triglyceride level were significantly decreased in *Lgr4^iKO* mice fed NCD (Fig. 2a–c). In *Lgr4^iKO* mice fed HFD, liver weight, plasma and hepatic triglyceride contents, and steatosis evidenced by H&E and oil red O staining were all decreased (Fig. 2d–f). These results suggest that deficiency of intestinal *Lgr4* protects mice from HFD-induced hepatic steatosis. Interestingly, hepatic genes relevant to lipogenesis, lipid transport, and β-oxidation remained largely unchanged (Supplementary Fig. 3). The only exception was the upregulation of *Ppara* (Supplementary Fig. 3). These results indicate that deficiency of intestinal *Lgr4* decreases hepatic lipid deposition likely via an extrahepatic mechanism.

### Deficiency of intestinal *Lgr4* decreases lipid absorption

The primary function of intestine is nutrient absorption. To assess the alteration of lipid absorption, we collected feces from mice, extracted total lipid, then measured fecal triglyceride levels. As shown in Fig. 3a, fecal triglyceride levels were significantly increased in *Lgr4^iKO* mice fed either NCD or HFD. Since HFD-feeding may alter lipid absorption, mice fed NCD were used in the following experiments. Plasma triglyceride levels and AUC of oral lipid tolerance test (OLTT) were substantially reduced in *Lgr4^iKO* mice after olive oil gavage (Fig. 3b). Oil red O staining showed that lipid droplets were remarkably decreased in the jejunum of *Lgr4^iKO* mice 2 h after gavage with 200 μL of olive oil (Fig. 3c). Examination of key intestinal lipid transporters revealed that both mRNA and protein levels of FATP4 and CD36 were substantially reduced (Fig. 3d–f). These results indicate that deficiency of intestinal

*Lgr4* decreases the expression of lipid transporters, leading to subsequent reduction in lipid absorption.

We next used MODE-K cells as an in vitro model to examine the effects of LGR4 on lipid absorption. Seventy-two hours after *Lgr4* siRNA treatment, mRNA and protein levels of LGR4 and FATP4 in MODE-K cells were substantially reduced (Fig. 3g, h). These alterations were associated with a reduction in the concentration of intracellular triglyceride in MODE-K cells treated with a mixture of oleic acid and palmitic acid (Fig. 3i, $P = 0.0548$). Consistently, intensity of BODIPY fluorescence was also reduced (Fig. 3j). Next, we verified the effect of *Lgr4* deletion on lipid uptake in intestinal organoids. The growth process of intestinal organoids was showed in Supplementary Fig. 4a. We found a significant decrease of *Fabp1* and *Fatp4* mRNA level (Supplementary Fig. 4b) and uptake of BODIPY (Supplementary Fig. 4c) in *Lgr4*-deficiency intestinal organoids. These results suggest that *Lgr4* knockdown decreases lipid uptake.

### Intestine-specific knockdown of *Lgr4* improves glucose tolerance

In addition to absorption of lipid, we examined whether deficiency of *Lgr4* would affect intestinal absorption of carbohydrate. After intraperitoneal injection of glucose, *Lgr4^iKO* mice exhibited improved glucose tolerance, particularly in the HFD group (Fig. 4a, b). Further, insulin resistance index (HOMA-IR) was decreased and insulin sensitivity index (HOMA-IS) increased in *Lgr4^iKO* mice fed HFD (Fig. 4c), suggesting that the increment in glucose tolerance may be attributed to improved peripheral insulin sensitivity. Interestingly, levels of blood glucose after oral glucose administration were not significantly altered compared with littermate control mice (Fig. 4d, e). The differential results of OGTT and IPGTT indicates that intestinal glucose absorption was increased in *Lgr4^iKO* mice. As shown in Fig. 4f, g, both mRNA and protein levels of GLUT2 were substantially elevated. These results suggest that deficiency of intestinal *Lgr4* increases glucose absorption via up-regulation of glucose transporter, GLUT2.

### Deficiency of intestinal *Lgr4* decreases enterocytes selective for long-chain fatty acid absorption while increasing carbohydrate-absorptive enterocytes

Intestinal epithelial populations are crucial for dietary lipid and carbohydrate absorption. To determine the mechanism underlying the decrease of lipid absorption and concurrent increase of carbohydrate absorption, we analyzed the cellular heterogeneity of intestinal epithelia using the single cell RNAseq. After quality control of data filtering, intestinal epithelial populations were re-clustered using Seurat. According to the reported marker genes[7], stem cells, TA cells, absorptive enterocytes, goblet cells, Paneth cells, enteroendocrine cells and tuft cells were defined (Fig. 5a, b). To further elucidate the order of differentiation between the epithelial populations, pseudotime analysis was analyzed and the cell markers of each population were projected on pseudotime axis. As shown in Fig. 5c, the differentiation of each population is well in line with the known order. These results confirm the accuracy for our definition of intestinal epithelial populations.

To investigate the cellular heterogeneity, absorptive enterocytes were re-clustered. Long-chain fatty acid-absorptive, and carbohydrate-absorptive enterocytes were defined by the expression of *Cd36* and *Fatp4*、*Glut2* and *Sglt1* respectively (Fig. 5d). The statistical analysis for the proportions of absorptive cells revealed a decrease in enterocytes selective for long-chain fatty acid absorption and an increase in enterocytes elective for carbohydrate absorption in intestine of *Lgr4^iKO* mice (Fig. 5e).

### LGR4 regulates differentiation of intestinal stem cells via Wnt and Notch signaling pathways

Absorptive enterocytes derive from the differentiation of intestinal stem cells. LGR4 has been found to be highly expressed in intestinal

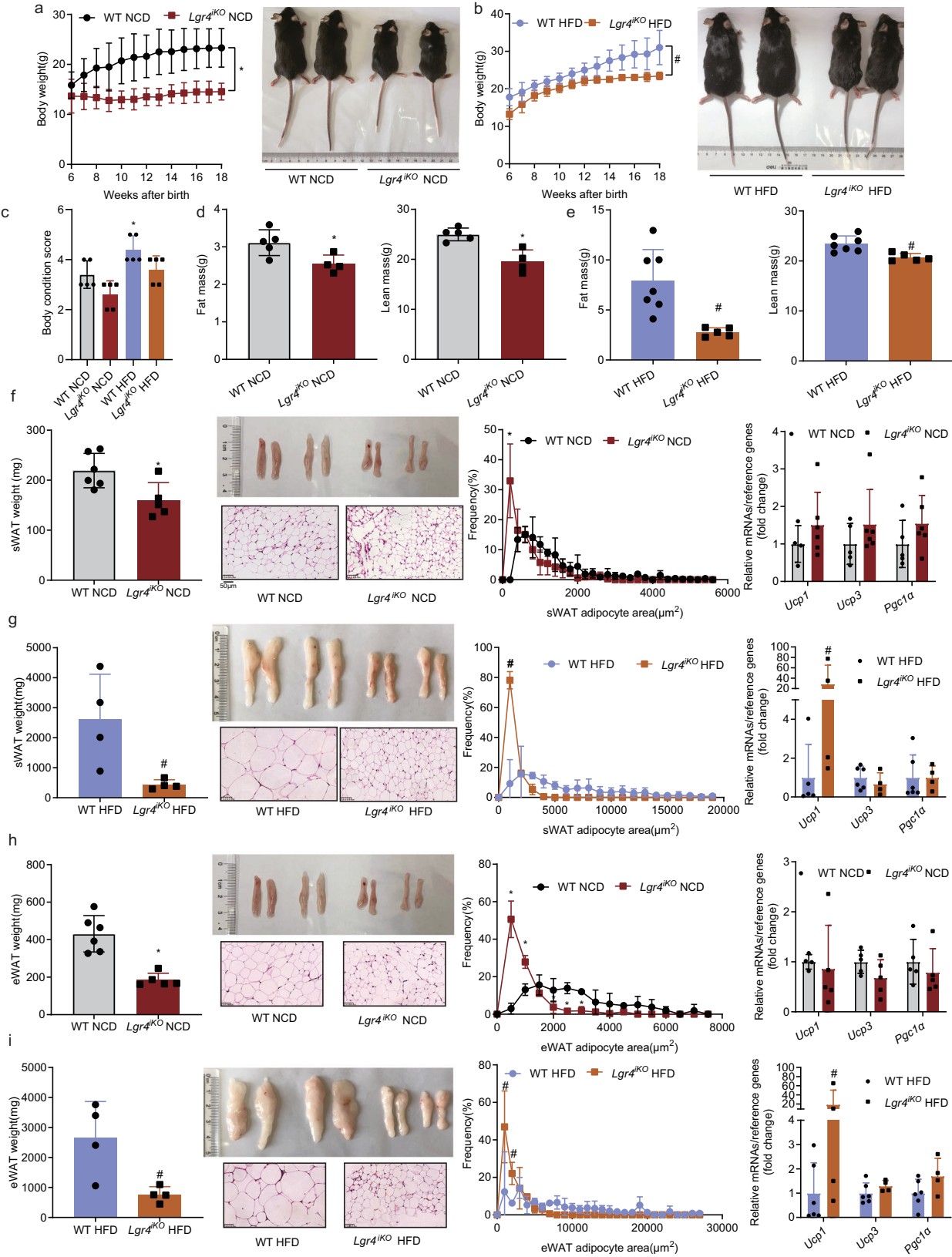

stem cells and TA cells[8]. To investigate the effect of intestinal LGR4 on stem cell differentiation, the proportions of each intestinal epithelial population was analyzed via single cell RNAseq. According to analysis of the number, the marker genes expression and specific staining, we found that deficiency of intestinal *Lgr4* increased the proportion of absorptive enterocytes and TA cells, while decreasing the proportion

of stem cells, goblet cells, Paneth cells, enteroendocrine cells and tuft cells (Supplementary Figs. 5, 6 and Fig. 6a). This observation was further confirmed by experiments using organoids derived from WT and *Lgr4* deficient mice (Supplementary Fig. 6l). Consistent with the decrement in the number of Paneth cells and Tuft cells, intestinal barrier disruption (Supplementary Fig. 7a–d) and alteration of the

**Fig. 1 | Deficiency of intestinal *Lgr4* protects mice from HFD-induced obesity.**
Six-week-old male *Lgr4^iKO* mice and littermates were fed normal chow diet (NCD) or 60% high fat diet (HFD) for 12 weeks. Results were expressed as mean ± SD and analyzed by the t-test (two-side). *P < 0.05 vs WT NCD. #P < 0.05 vs WT HFD. *n* = 4–11. **a**, **b** Body weight and body size of mice fed NCD or HFD. *n* = 11 for WT NCD, 5 for *Lgr4^iKO* NCD, 7 for WT HFD, and 5 for *Lgr4^iKO* HFD. WT NCD vs. *Lgr4^iKO* NCD: *P* = 0.0003. WT HFD vs. *Lgr4^iKO* HFD: *P* = 0.005. **c** Body condition score (BCS) of mice fed NCD or HFD. *n* = 5. *P* = 0.0476. **d**, **e** Fat mass and lean mass of mice fed

NCD or HFD. *n* = 5 for WT NCD, 4 for *Lgr4^iKO* NCD, 7 for WT HFD, and 5 for *Lgr4^iKO* HFD. **d** Fat mass: *P = 0.0268. Lean mass: *P = 0.0027. **e** Fat mass: #P = 0.0048. Lean mass: #P = 0.006. **f**–**i** Fat mass, H&E staining, adipocyte size and mRNA levels of beigeing marker genes in sWAT (*Ucp1, Ucp3 and Pgc1a*) of mice fed either NCD (**f**: *n* = 5–6, *P* = 0.0196) or HFD (**g**: *n* = 4–6, #P = 0.0279 for sWAT weight, #P <0.0001 for frequency, #P = 0.0115 for *Ucp1*), as well as in eWAT of mice fed NCD (**h**, *n* = 4–6, *P* = 0.0005) or HFD (**i**, *n* = 4–6, #P = 0.0222 for eWAT weight, #P = 0.047 for *Ucp1*).

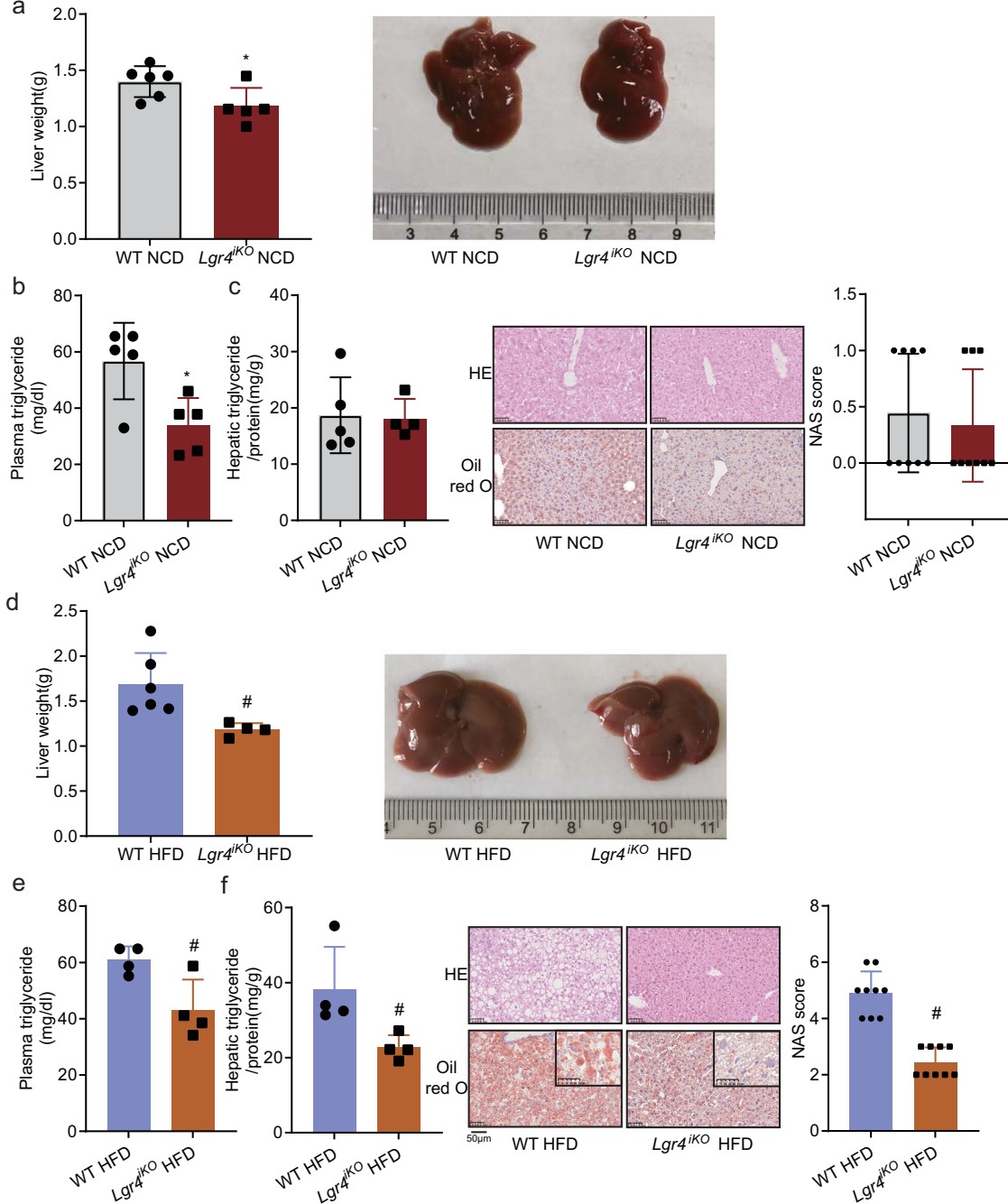

**Fig. 2 | Deficiency of intestinal *Lgr4* protects mice from HFD-induced hepatic steatosis.** Six-week-old male *Lgr4^iKO* mice and littermates were fed normal chow diet (NCD) or 60% high fat diet (HFD) for 12 weeks. Results were expressed as mean ± SD and analyzed by the t-test (two-side). *P < 0.05 vs WT NCD. #P < 0.05 vs WT HFD. **a** Liver weight and liver size of NCD-fed mice. *n* = 6 for WT NCD and 5 for *Lgr4^iKO* NCD. *P* = 0.0385. **b** Plasma triglyceride levels of NCD-fed mice. *n* = 5.

*P* = 0.0157. **c** Triglyceride contents and steatosis in liver of NCD-fed mice. *n* = 5 for WT NCD and 4 for *Lgr4^iKO* NCD. **d** Liver weight and liver size of HFD-fed mice. *n* = 6 for WT NCD and 4 for *Lgr4^iKO* NCD. #P = 0.024. **e** Plasma triglyceride levels of HFD-fed mice. *n* = 4. #P = 0.0236. **f** Triglyceride contents and steatosis in liver of HFD-fed mice. *n* = 4. #P = 0.0388 for hepatic triglyceride, #P < 0.0001 for NAS score.

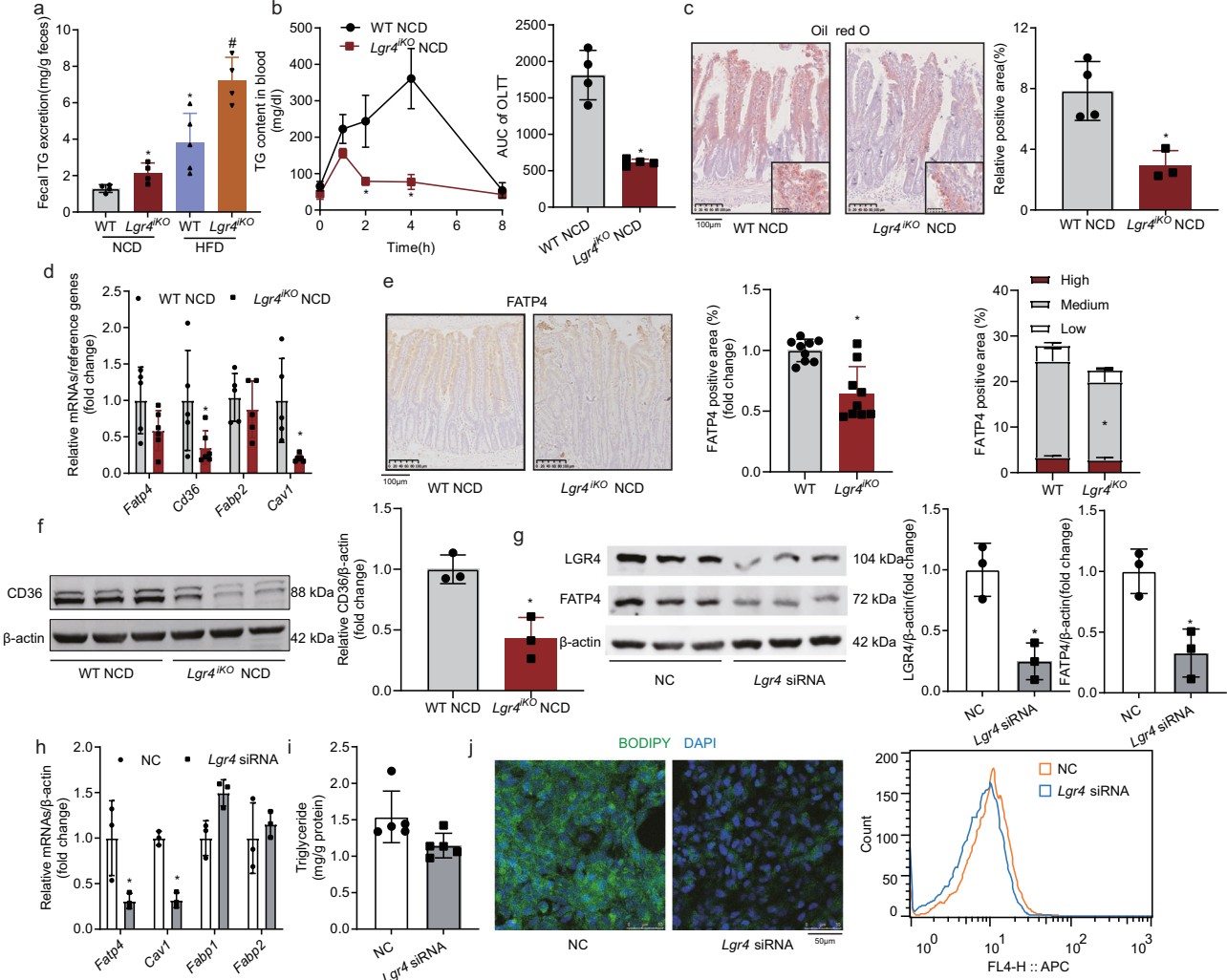

**Fig. 3 | Deficiency of intestinal _Lgr4_ decreases lipid absorption. a–f** Six-week-old male _Lgr4^{iKO}_ mice and littermates were fed normal chow diet (NCD) or 60% high fat diet (HFD) for 12 weeks. Results were expressed as mean ± SD and analyzed by the t-test or one-way ANOVA. *$P < 0.05$ vs WT NCD. #$P < 0.05$ vs WT HFD. $n = 3$–6. **a** Fecal triglyceride levels. $n = 4$ for WT NCD, 4 for _Lgr4^{iKO}_ NCD, 5 for WT HFD, and 4 for _Lgr4^{iKO}_ HFD. WT NCD vs. _Lgr4^{iKO}_ NCD: *$P = 0.0296$. WT NCD vs. WT HFD: *$P = 0.0185$. WT HFD vs. _Lgr4^{iKO}_ HFD: #$P = 0.011$. **b** Levels of circulating triglyceride and the area under curve in response to oral administration of olive oil in NCD-fed mice. $n = 4$. *$P = 0.0004$ for AUC. **c** Oil red O staining of intestine 2 h after olive oil gavage and quantitative analysis. $n = 4$ for WT NCD and 3 for _Lgr4^{iKO}_ NCD. *$P = 0.0107$. **d** mRNA levels of lipid absorption markers (_Fatp4, Cd36 and Fabp2_) in intestine of NCD-fed mice. $n = 5$ for WT NCD and 5–6 for _Lgr4^{iKO}_ NCD. *$P = 0.0483$ for _Cd36_, *$P = 0.0104$ for _Cav1_. **e** Immunohistochemical staining of FATP4 in

intestine and quantification of positive area. $n = 9$. *$P = 0.0004$(left) and 0.0075(right). **f** Western blot and quantification of CD36 protein levels. β-actin was used as internal control. $n = 3$. *$P = 0.0093$. **g–j** The mouse small intestinal epithelial cell line MODE-K cells were transfected with _Lgr4_ siRNA for 48 h. **g** Western blot and quantification of LGR4 and FATP4 protein levels. β-actin was used as loading control. $n = 3$. *$P = 0.008$ for LGR4, *$P = 0.0124$ for FATP4. **h** mRNA levels of lipid absorption markers (_Fatp4, Cav1, Fabp1 and Fabp2_) in MODE-K cells. _β-actin_ was used as a reference gene. $n = 3$. *$P = 0.0077$ for _Fatp4_, *$P = 0.0085$ for _Cav1_. **i** The triglyceride level in MODE-K cells treated with mixture of oleic acid (0.6 mmol/l) and palmitic acid (0.2 mmol/l). $n = 5$. **j** Cells were treated with BODIPY-$C_{12}$ long-chain fatty acid for 2 h and observed under microscope and the uptake of BODIPY-$C_{12}$ long-chain fatty acid in MODE-K cells by flow cytometry.

microbiota (Supplementary Fig. 8) were observed in _Lgr4^{iKO}_ mice. In addition, _Lgr4_ deficiency supressing the apoptosis indicates that the survival time of enterocytes is increased (Supplementary Fig. 7e).

Since the differentiation of intestinal stem cells into absorptive and secretory progenitors is regulated by Notch and Wnt signaling respectively[9,10], we next examined the alteration of these two signaling pathways in _Lgr4^{iKO}_ mice. As shown in Fig. 6b and c, cells highly expressing the absorptive progenitor genes (_Ccnb1, Cdc20, Cenpa, Cdkn3, Ube2c, Aurka_ and _Ccna2_) substantially increased, whereas cells highly expressing the secretory progenitor gene (_Dll1_) decreased. Consistently, mRNA levels of absorptive progenitor marker and secretory progenitor marker genes were strikingly increased and decreased respectively (Fig. 6d) in intestine of _Lgr4^{iKO}_ mice. These results indicate that deficiency of intestinal _Lgr4_ increases proportion

of absorptive progenitors while concurrently decreases proportion of secretory progenitors.

The differentiation of secretory progenitors is activated by Wnt-β-catenin signaling, whereas the differentiation of absorptive progenitors depends on Notch signaling[9,11–14]. Consistently, cells highly expressing the Wnt target genes such as _Sox9, Sox4, Ascl2, Myc, Ube2c_, and _Lgr5_, and the UMI value of these genes were decreased in _Lgr4^{iKO}_ mice (Fig. 6e). Further, nuclear translocalization of β-catenin was decreased in intestine epithelia of _Lgr4^{iKO}_ mice (Fig. 6f). mRNA levels of _Math1_, the downstream target of β-catenin was decreased (Fig. 6g). On the other hand, cells highly expressing Notch related genes such as _Notch1, Notch4, Dtx1, Dtx3, Dll4_, and _Mfng_ and the UMI value of these genes were significantly increased (Fig. 6h). mRNA and protein levels of HES1, the downstream target of Notch was increased (Fig. 6i). As

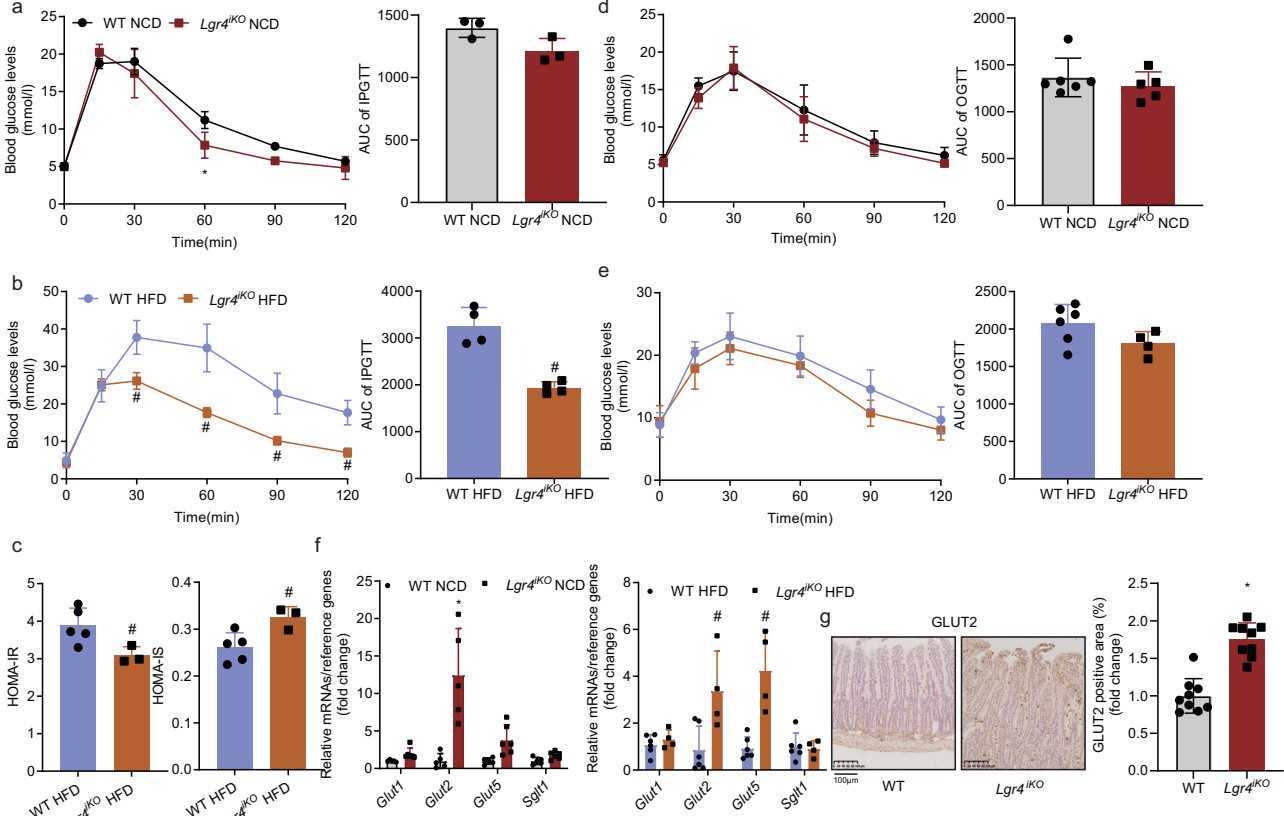

**Fig. 4 | Intestine-specific *Lgr4* knockout improves glucose tolerance.** Six-week-old male *Lgr4^iKO* mice and littermates were fed normal chow diet (NCD) or 60% high fat diet (HFD) for 12 weeks. Results were expressed as mean ± SD. *$P < 0.05$ vs WT NCD. #$P < 0.05$ vs WT HFD. $n = 3$–6. **a**, **b** Intraperitoneal glucose tolerance test and the area under curve of mice fed NCD or HFD. $n = 3$ for WT NCD, 3 for *Lgr4^iKO* NCD, 4 for WT HFD, and 4 for *Lgr4^iKO* HFD. Statistical analysis by two-way ANOVA with Šídák's multiple comparisons test for IPGTT. #$P = 0.0007$. **c** Homeostatic model assessment of insulin resistance (HOMA-IR) and homeostatic model assessment of insulin sensitivity (HOMA-IS) of HFD-fed mice. HOMA-IR = plasma insulin (μU) × glucose (mmol/l)/22.5. HOMA-IS = 1/HOMA-IR. $n = 5$ for WT HFD, and 3 for *Lgr4^iKO* HFD. Student's t test (two-side) was used for un-paired analysis. #$P = 0.0371$ for HOMA-IR. #$P = 0.0234$ for HOMA-IS. **d**, **e** Oral glucose tolerance test and the area under curve in mice fed NCD or HFD. $n = 6$ for WT NCD, 5 for *Lgr4^iKO* NCD, 6 for WT HFD, and 4 for *Lgr4^iKO* HFD. **f** mRNA levels of carbohydrate absorption markers (*Glut1, Glut2, Glut5 and Sglt1*) in intestine of mice fed NCD or HFD. $n = 5$ for WT NCD, 5–6 for *Lgr4^iKO* NCD, 6 for WT HFD, and 4 for *Lgr4^iKO* HFD. Statistical analysis by two-way ANOVA with Šídák's multiple comparisons test. *$P < 0.0001$. #$P = 0.0007$ for *Glut2* and #$P < 0.0001$ for *Glut5*. **g** Immunohistochemical staining of GLUT2 in intestine and quantification of positive area. $n = 9$.

shown in Fig. 6j, activation of Notch signaling by VPA significantly suppresses the Wnt signaling evidenced by the decrement of its relevant target genes. This observation suggests a link between Wnt and Notch signaling. In addition, inhibition of Wnt signaling substantially decreases expression of lipid uptake genes *Fatp4* in MODE-K cells (Fig. 6k). This observation suggests that inhibition of Wnt signaling may decrease lipid absorption. These results indicate that LGR4 regulates the differential differentiation of intestinal stem cells into absorptive and secretory cells via Notch and Wnt signaling pathways.

Next, we further explored the relationship between LGR4 and Notch signaling in our research. In MODE-K cells, we quantified the mRNA levels of genes in Notch signaling, and found that levels of *Psen1*, *Aph1a* and *Nedd4* were significantly changed (Supplementary Fig. 9a). We then further confirmed the increment of PSEN1 mRNA (Supplementary Fig. 9b) and protein levels (Supplementary Fig. 9c) in IEC6 cells. PSEN1 is an active component of γ-secretase, which is responsible for the cleavage of Notch and next-step function of NICD. These observations suggest that deficiency of LGR4 enhances the expression of *Psen1* and thus stimulating the function of Notch signaling. The canonical pathway of LGR4 function is mediated by the nuclear translocation of β-catenin and then the transcriptional regulation of TCF/LEF transcription factor family. Using UCSC Genome Browser Home and JASPAR database, we predicted that TCF7L2 may

bind to the promoter region of *Psen1*. Therefore, we constructed pcDNA3.4-Tcf7l2 and pGL3-Psen1 promoter plasmids and cotransfected them with pRL-TK into 293T cells. We found that *Psen1* relative luciferase activity was significantly reduced in condition of *Tcf7l2* overexpression (Supplementary Fig. 9d). In organoids, *Lgr4* deficiency increased *Psen1* mRNA level (Supplementary Fig. 9e). Suppression of Wnt signaling by IWR-1 and activation of Notch using VPA mimicked, to some extent, the effects of *Lgr4* deficiency on differentiation of intestinal stem cells to lipid absorptive cells (Supplementary Fig. 9f and Supplementary Fig. 9g). Further, suppression of Notch signaling using DAPT attenuated the differentiation of intestinal stem cells into lipid absorptive cells (Supplementary Fig. 9h–j). Together, these results suggest that deficiency of LGR4 may activate Notch signaling via β-catenin and TCF7L2 mediated transcriptional regulation of *Psen1*.

## Discussion

LGR4 is abundantly expressed in digestive organs[15,16]. Previous studies have been focused on its role in intestinal development[17–19]. Our studies extend the physiological functions of intestinal LGR4 to modulation of lipid absorption and lipid homeostasis. Using the genetic approach to knockdown the *Lgr4* gene specifically in intestinal epithelia, we demonstrate the metabolic benefit induced by deficiency of intestinal *Lgr4*. Knockdown of *Lgr4* in intestinal epithelia rendered the

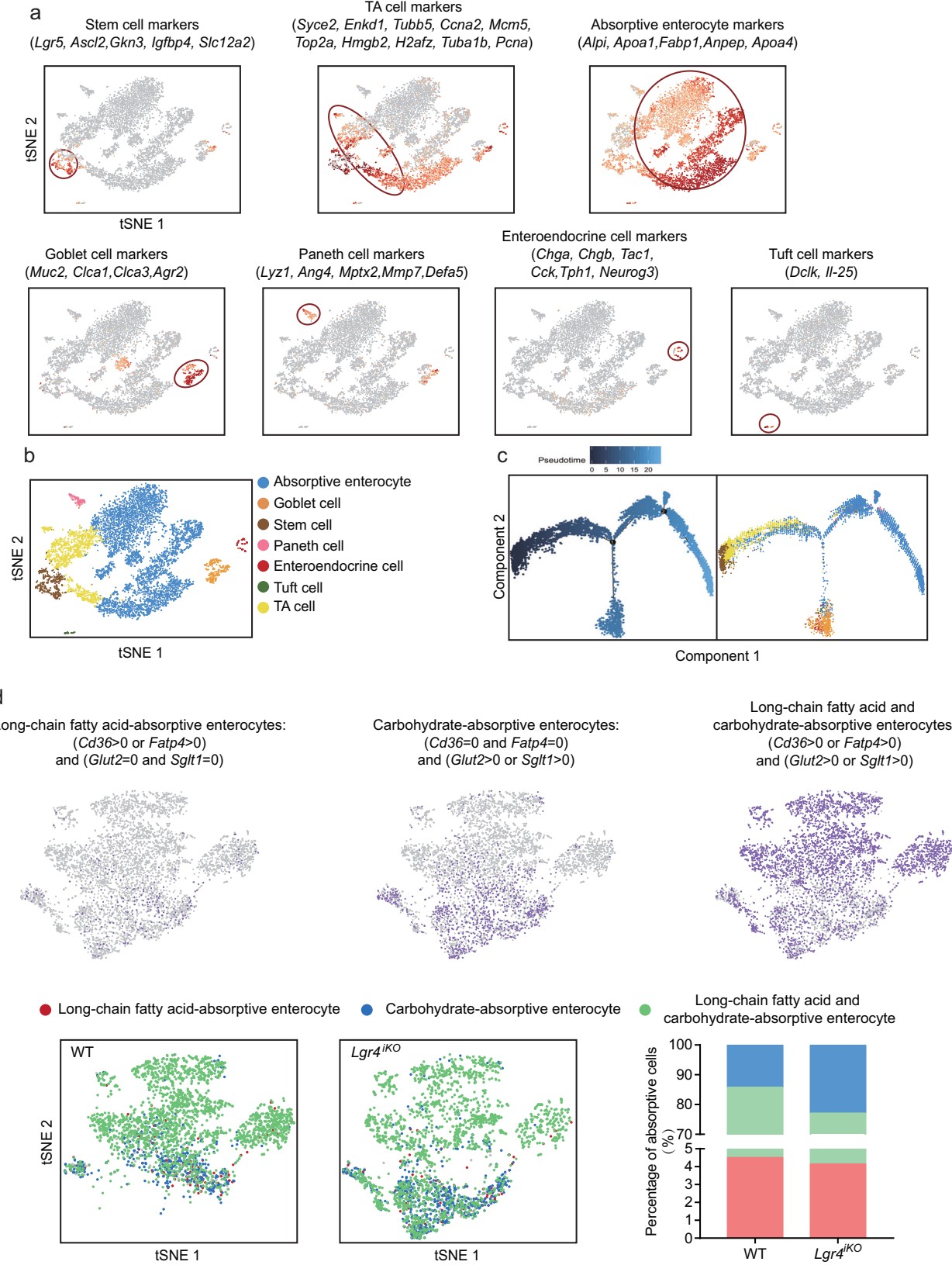

**Fig. 5 | The effect of LGR4 on the heterogeneity of intestinal absorptive cells.** Six-week-old male *Lgr4iKO* mice and littermates were fed normal chow diet (NCD) for 12 weeks. Single cell RNA sequencing was used to obtain intestinal epithelium single cell transcriptome data from 18-week-old *Lgr4iKO* mice and littermates. *n* = 3. **a** t-SNE plot showing stem cell, TA cell, absorptive cell, goblet cell, paneth cell, enteroendocrine cell and tuft cell marker genes expression in intestinal epithelium. **b** Defining cell populations with marker gene expression. **c** Pseudotime ordering on intestinal epithelium cells. **d** Re-clustering absorptive cells. Enterocytes selective for absorption of long-chain fatty acid, carbohydrate, or both were defined by the expression of *Cd36、Fatp4、Glut2* and *Sglt1*. **e** The proportions of long-chain fatty acid-absorptive enterocytes, carbohydrate-absorptive enterocytes, and long-chain fatty acid and carbohydrate-absorptive enterocytes.

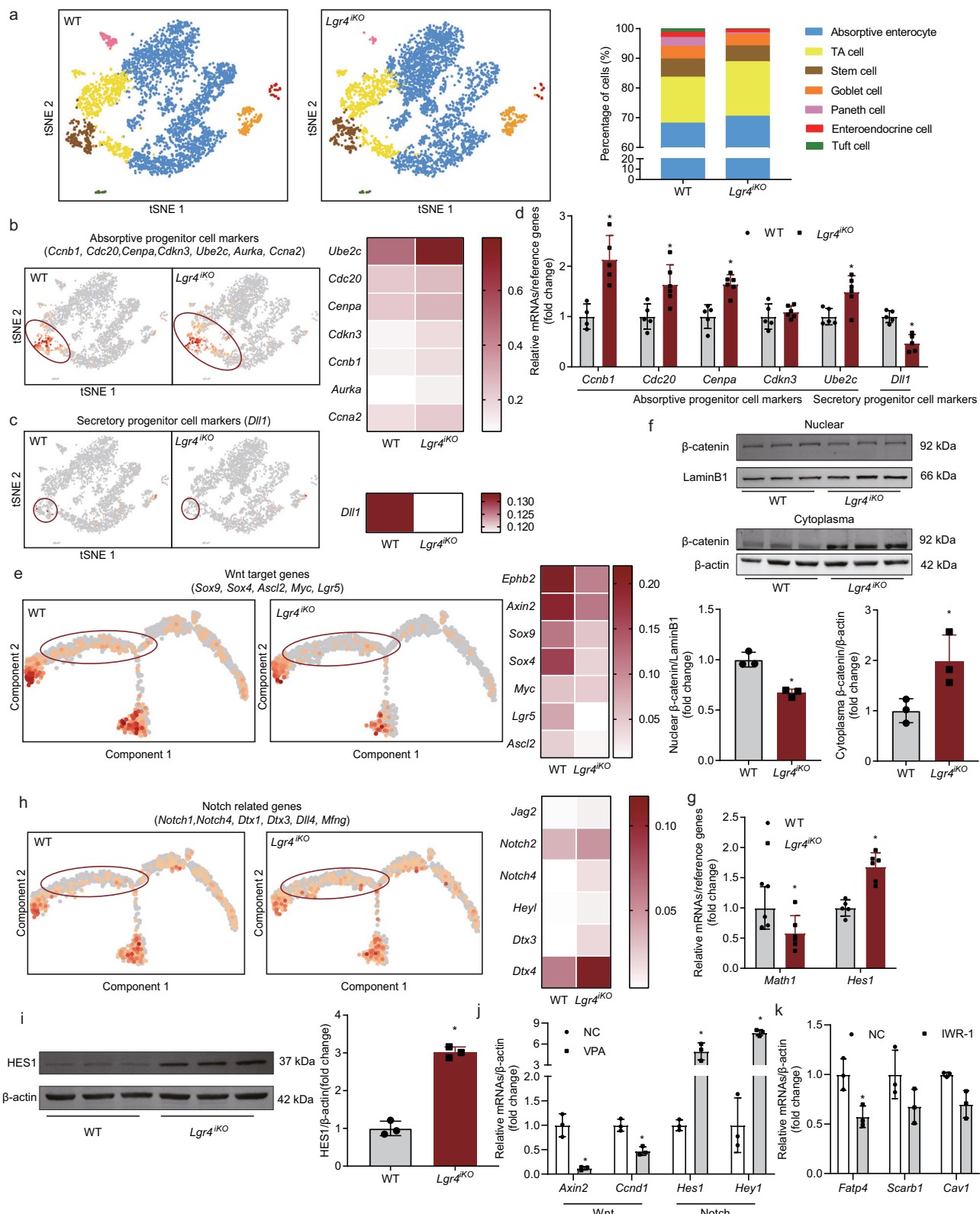

mice resistant to HFD-induced obesity and its related metabolic disorders. Significant reduction in adiposity and liver steatosis was observed in *Lgr4iKO* mice. The improvement in lipid metabolism appears not to be attributed to energy expenditure because coldexposure induced thermogenesis, physical activity and respiratory quotient were not altered. Rather, it is the reduction in lipid absorption that protects *Lgr4iKO* mice from HFD-induced lipid disorders. Shortly

after administration of lipid, plasma levels of triglyceride and lipid droplets inside the intestinal epithelia were substantially lower in *Lgr4iKO* mice relevant to the wild type littermates. On the other hand, fecal levels of lipid were increased in the *Lgr4iKO* transgenes. All these observations suggest the significant reduction of lipid absorption in *Lgr4iKO* mice. The metabolic benefit of the decreased lipid absorption is significant. Mice with intestinal epithelial knockout of *Lgr4* were

**Fig. 6 | LGR4 regulates differentiation of ISCs via Wnt and Notch signaling pathways. a–i** Six-week-old male *Lgr4^iKO* mice and littermates were fed normal chow diet for 12 weeks. Single cell RNA sequencing was used to obtain intestinal epithelium single cell transcriptome data from 18-week-old *Lgr4^iKO* mice and littermates. *n* = 3. \**P* < 0.05 vs WT. **a** The proportions of stem cells, TA cells, absorptive cells, goblet cells, paneth cells, enteroendocrine cells and tuft cells. **b** t-SNE plot showing absorptive progenitor cell marker genes expression (left) and heatmap showing UMI value (right). The transcription expression levels were calculated as UMI value. **c** t-SNE plot showing secretory progenitor cell marker genes expression (left) and heatmap showing UMI value of *Dll1* (right). **d** mRNA levels of absorptive progenitor cell and secretory progenitor cell markers in intestine of NCD-fed mice. *n* = 4–5 for WT, and 5–6 for *Lgr4^iKO*. Results were expressed as mean ± SD. **e** Pseudotime showing Wnt target genes expression (left) and heatmap showing UMI value (right). **f** Western blotting detecting nuclear and cytosol levels of β-

catenin protein. The relative expression level was quantified using Image J software. Results were expressed as mean ± SD. **g** mRNA levels of downstream genes of the Notch signaling pathway (*Math1* and *Hes1*) detected by RT-qPCR. *n* = 5 for WT, and 6 for *Lgr4^iKO*. Results were expressed as mean ± SD. **h** Pseudotime showing Notch related genes expression (left) and heatmap showing UMI value (right). **i** Western blot and quantification of HES1 protein levels. β-actin was used as loading control. Results were expressed as mean ± SD. **j** Wnt and Notch related genes of the mouse small intestinal epithelial cell line MODE-K cells treated with the Notch activator VPA (P4543-25G, Sigma). \**P* < 0.05 vs NC. *n* = 3. Statistical analysis by two-way ANOVA with Šídák's multiple comparisons test. \**P* = 0.0001 for *Axin2*, \**P* = 0.0035 for *Ccnd1*, \**P* = 0.0003 for *Hes1*, \**P* < 0.0001 for *Hey1*. **k** mRNA levels of lipid absorption markers of MODE-K cells treated with the Wnt inhibitor IWR-1 (HY-12238, MedChemExpress). \**P* < 0.05 vs NC. *n* = 3. Statistical analysis by two-way ANOVA with Šídák's multiple comparisons test. \**P* = 0.0178.

resistant to HFD-induced adiposity, liver steatosis and deterioration of glucose tolerance. Interestingly, although intraperitoneal glucose tolerance improved significantly, oral glucose tolerance demonstrated no alteration. This result indicates a substantial increment of glucose absorption in the intestine of *Lgr4^iKO*.

The observation that deficiency of intestinal *Lgr4* reduces intestinal lipid absorption while increasing glucose absorption suggests the functional significance of intestinal LGR4 in determining the selective absorption of nutrients. Each individual intestinal epithelial cell is classically considered to be capable of absorbing multiple nutrients such as glucose, fatty acid and amino acid without selection. Our study provides evidence challenging this concept. Using *Cd36* and *Fatp4*, and *Glut2* and *Sglt1* to define enterocytes selective for absorption of long-chain fatty acid or carbohydrate, we have revealed the heterogeneity of intestinal absorptive cells. Three distinct populations of absorptive cells are present in intestinal epithelia, including cells selective for absorption of long-chain fatty acid, carbohydrate, or both. Interestingly, the decrement in enterocytes selective for absorption of long-chain fatty acid was associated with the concurrent increment in enterocytes selective for absorption of carbohydrate. The physiological significance of this concurrent alteration is unknown but may be related to protection against the uncontrolled insufficiency of energy absorption. Our study thus provides evidence supporting the cellular heterogeneity of absorptive enterocytes. Consistently, a recent spatial transcriptomics study has found that the expression of genes related to carbohydrate and lipid transport in absorptive cells differs in spatial distribution location[20]. This observation suggests the spatial heterogeneity in absorptive enterocytes absorbing various nutrients.

Global knockout of *Lgr4* has been reported to block the terminal differentiation of Paneth cells only[4], while there is no evidence for its effect on the differentiation of absorptive, enteroendocrine, and goblet-cell lineages. Surprisingly, our studies reveal that intestine-specific deletion of *Lgr4* reduced proportion of enteroendocrine, tuft and goblet cells in addition to Paneth cells. These results indicate that LGR4 is crucial for the differentiation of secretory enterocytes. LGR4 binds directly to endogenous ligands such as RSPO1/2/3/4, Norrin, Nidogen-2, and RANKL[21–25]. Downstream signaling mediated by LGR4 primarily includes cAMP/PKA, Wnt/β-catenin, Gαq/GSK3β and Gαq/PKCα signaling pathways. In addition, LGR4 can regulate AKT, extracellular signal-regulated kinase (ERK), and nuclear factor kappa-B (NF-κB) pathways[26]. However, how these signaling pathways regulate intestine epithelium remains unclear, and needs further exploration. Previous studies have demonstrated that differentiation of secretory occurs via activation of Wnt[9,10]. Consistently, deficiency of intestinal *Lgr4* inhibits Wnt signaling in our study. On the other hand, deficiency of intestinal *Lgr4* activates Notch signaling, leading to subsequent increment in the proportion of absorptive enterocytes. This observation is in line with previous report demonstrating that activation of Notch signaling is required for the differentiation of absorptive enterocytes[11–14]. As a limit of this study, the molecular mechanism by

which LGR4 determines the selective differentiation of absorptive enterocytes is unknown. Our data suggest that deficiency of LGR4 may activate Notch signaling via β-catenin and TCF7L2 mediated transcriptional regulation of *Psen1*. Further experiment should explore whether the interaction between Notch and Wnt signaling pathways requires *Psen1* using the genetic approach.

In summary, our study demonstrates that ablation of *Lgr4* gene in intestinal epithelium affects stem cell differentiation via the activation of Notch and concurrent suppression of Wnt signaling pathway. This bias of stem cells differentiation results in reduced proportion of lipid-absorptive enterocytes, leading to decrement of lipid absorption and subsequent improvement in glucose and lipid metabolism (Fig. 7). Our results reveal the cellular heterogeneity of intestinal absorptive cells and the crucial role of intestinal LGR4 in controlling the differentiation of enterocytes selective for absorption of lipid. Our study thus suggests that targeting intestinal LGR4 may provide a potential strategy for the intervention of obesity and liver steatosis.

## Methods

### Materials

Rabbit anti-LGR4 was purchased from Abcam (ab75501, Cambridge, MA). Rabbit anti-CD36 (18836-1-AP) was purchased from Proteintech (Chicago, USA). Rabbit anti-GLUT2 (A9843), rabbit anti- Presenilin 1(A2187), rabbit anti-LYZ (A13511) and rabbit anti-HES1 (A0925) were from ABclonal (Wuhan, China). Rabbit anti-FATP4 was obtained from Abmart (T57249, Shanghai, China). Rabbit anti-β-catenin was from Cell Signaling Technology (8480, Danvers, MA, USA). Rabbit anti-LaminB1 (12987-1-AP) and mouse anti-β-actin (6009-1-Ig) were purchased from Proteintech (Chicago, USA). Rabbit anti-Ki67 was purchased from Abcam (ab15580, Cambridge, United Kingdom). Rabbit anti-OLFM4 was obtained from Cell Signaling Technology (39141, Danvers, MA, USA). *Lgr4* siRNA (5′-GGACUUAUCUUAUAACGAUUTT-3′) was synthesized by Synbio Technologies (Suzhou, China). RNAi in vitro transfection reagent was obtained from D-Nano Therapeutics (DN001-10, Beijing, China). TUNEL Apoptosis Assay Kit was purchased from Beyotime Biotechnology (Beyotime, Shanghai, China).

### Animals and treatment

All experiments were conducted in strict accordance with the Guide for the Care and Use of Laboratory Animals prepared by the National Academy of Sciences (NIH publication 86-23, revised 1985). Experimental protocols were approved by the Animal Care and Use Committee of Peking University. *Villin-Cre* mice were bred with *Lgr4^flox/flox* mice (from Helmholtz Zentrum, Germany) to generate *Lgr4^iKO* mice, within which *Lgr4* was specifically knocked out in intestinal epithelium. Six-week-old male *Lgr4^iKO* mice and wild type (WT) littermates were fed normal chow diet (10 kcal% fat；D12450H；Research Diets) or high fat diet (60 kcal% fat；D12492；Research Diets) for 12 weeks. Animals were housed in a standard environment (22 ± 2 °C, humidity at 50 ± 15%) with 12 h light and 12 h dark cycle. Food and water were freely

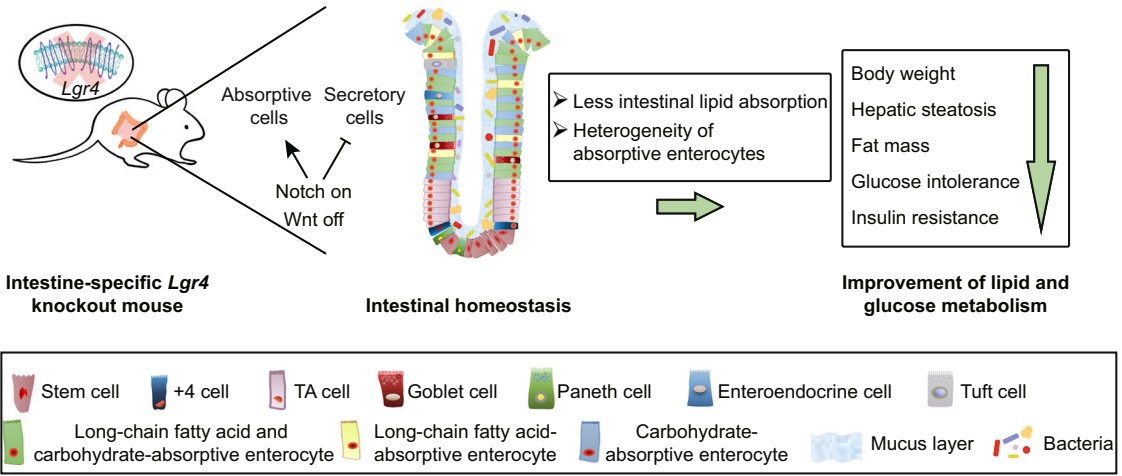

**Fig. 7 | Graphic highlight of findings.** Deficiency of intestinal *Lgr4* reduces enterocytes selective for absorption of long-chain fatty acid, leading to reduction in lipid absorption and subsequent metabolic benefit.

accessible except for the fasting experiment. At the end of the experiment, following tissue samples were harvested: plasma, intestine, liver, epididymal white adipose tissue (eWAT), and subcutaneous white adipose tissue (sWAT).

### Tissue sample preparation and histological analysis

The dissected tissues were fixed with 4% paraformaldehyde in PBS for 24 h at 4 °C and stored in 20% sucrose phosphate buffer. Samples were embedded in paraffin or OTC and sectioned at a 3μm thickness (or 10μm for oil red O staining). H&E, immunohistochemical or oil red O staining were performed following general protocols. For the detection of intestinal Alkaline Phosphatase (ALPI), staining was performed according to the manufacturer's instructions (G1480, Solarbio, China). Periodic Acid Schiff (PAS) staining was performed according to the manufacturer's instructions (C0142, Beyotime, China). For Grimelius technique to stain enteroendocrine cells, deparaffinized and rehydrated samples were incubated in a preheated silver solution at 60 °C for 3 h. Afterwards, sections were washed in distilled water. then incubated in the 45 °C preheated reducing solution (1 g hydroquinone +2.5 g anhydrous sodium sulfite +100 ml ultra-pure water) for 1 h. Afterward, sections were washed in distilled water, dehydrated, transparentized and inspected by light microscope.

### Oral glucose tolerance tests (OGTT) and Intraperitoneal glucose tolerance test (IPGTT)

Mice fasted for 16 h were orally gavaged (for OGTT) or intraperitoneally injected (for IPGTT) with glucose at a dose of 3 g/kg body weight. Blood was collected from the incision at the tip of the tail at 0, 15, 30, 60, 90, and 120 min after glucose administration, and the glucose concentration was immediately measured by a glucometer (Accu-Chek Active, Roche, Germany).

### Oral lipid tolerance test (OLTT)

Mice were fasted for 16 h before oral administration of olive oil (200 μL). Blood was collected from inner canthus at 0, 1, 2, 4, and 8 h after olive oil administration, and serum triglyceride were measured by colorimetry.

### Western blot and quantitative RT-PCR

Protein was quantitated and loaded onto an SDS-PAGE gel and transferred to a PVDF membrane. Membrane was blocked with 5% nonfat dry milk in TBST at room temperature for 1 h, then incubated with the primary antibody overnight at 4 °C. IRDye-labeled secondary antibodies were used to detect specific reactions and visualized with the Odyssey infrared imaging system. The relative level of protein was assessed using Image J software.

RNA was isolated from tissues using Trizol, followed by reverse transcription. Quantitative real time-PCR was performed using SYBR green in Agilent AriaMx real-time PCR system. Supplementary Table 1 showed the primer sequences involved in this study. mRNA levels were normalized to the geometric mean value of reference genes (*Hprt*, *Rpl32* and *Tbp*) or *β-actin*.

### Cell culture and siRNA transfection

MODE-K cells were cultured in a humid atmosphere (5% $CO_2$) using RPMI-1640 culture medium supplemented with 10% FBS and 1% penicillin/streptomycin at 37 °C. IEC-6 cells and 293T cells were cultured in a humid atmosphere (5% $CO_2$) using DMEM culture medium supplemented with 10% FBS and 1% penicillin/streptomycin at 37 °C. Cells were seeded in 12-well plate and grown until 70% confluency before transfection. Cells were washed with sterile phosphate-buffered saline (PBS) before transfection, then incubated with 50 nM *Lgr4* siRNA or non-targeting Control siRNA (Synbio Technologies, Suzhou, China) for 48 h in a volume of 1 ml/well. Cells were then treated with mixture of oleic acid (0.6 mmol/l) and palmitic acid (0.2 mmol/l) or BODIPY-$C_{12}$ long-chain fatty acid (Invitrogen, Carlsbad, CA, USA) before harvesting for lipid absorption analysis.

### Isolation of intestinal organoids

Mice were sacrificed, and then a proper length of intestine was taken out and washed for several times. Split the intestinal cavity and scrape the villus with a blade. Cut the intestine into pieces and incubate them in PBS containing 5 mM EDTA for 30 min. Vortex and filter through a cell sieve with a diameter of 70 μm to obtain crypts. Centrifuge and resuspend the crypts with the mixture of IntestiCult™ Organoid Growth Medium (06000, STEMCELL Technologies, Canada) and Matrigel (356234, Corning, USA). Seed the mixture in 24-well plate and put it in the cell incubator for 20 min to allow Matrigel to solidify, and then add appropriate amount of OGM into the plate and culture in the cell incubator.

### Dual luciferase reporter gene assay

Mouse *Tcf7l2* CDS region was cloned to pcDNA3.4 vector and 1440bp-1930bp upstream mouse *Psen1* gene region (predicted binding site, Chromosome 12, NC_000078.7 (83732996. 83733555)) was cloned to pGL3-basic vector. Cotransfected the plasmids together with pRL-TK (E2241, Promega, USA) into 293T cells. Harvest the cells after 24 h and conducted dual luciferase reporter gene assay using a kit (KGAF040, KeyGEN BioTECH, China) following the manufacturer's instructions.

## Preparation of single-cell suspension of intestinal epithelium

The intestine was dissected longitudinally, minced, washed with PBS, then digested with 10 mmol/l EDTA on ice for 30 min. Cell suspension was filtered through 70 μm nylon mesh, centrifuged at 800 rpm for 5 min, then incubated with pre-warmed single cell digest containing dispase (1.67 U/ml) and DNaseI (0.01 mg/ml) at 37 °C. Cell suspension was then washed with PBS and filtered through 70 μm nylon mesh to remove the cell aggregates. Subsequently, cell suspension was centrifuged at 1000 rpm for 5 min at 4 °C. Cells were collected, and resuspended in PBS. Each sample was a pool of cells from the intestinal epithelium of 3 mice.

## Single-cell RNA sequencing

Single cell capture, library preparation, sequencing and data analysis were performed by Capitalbio Technology Corporation (Beijing, China). Single cell capture and library preparation were performed following general protocols. For Seurat pipeline, cells whose gene number was less than 200, or gene number ranked in the top 1%, or mitochondrial gene ratio was more than 25% were regarded as abnormal and filtered out. Dimensionality reduction was performed using PCA, and visualization was realized by TSNE. Cells were defined based on the marker genes of each population as reported[7]. Single-cell trajectories were built with Monocle (R package) that introduced pseudotime. Genes were filtered by the following criteria: Expressed in more than 10 cells; The average expression value was greater than 0.1; Qval was less than 0.01 in different analysis.

## Gut microbiota analysis by 16S rRNA gene sequencing

Total genome DNA from samples harvested from ileocecal contents was extracted using CTAB/SDS method. Following 16s rRNA V3-V4 region amplification, PCR products quantification and qualification, DNA library was constructed using the TruSeq® DNA PCR-Free Sample Preparation Kit. Sequencing libraries were generated using TruSeq® DNA PCR-Free Sample Preparation Kit (Illumina, USA) following manufacturer's recommendations and index codes were added. The library quality was assessed on the Qubit@ 2.0 Fluorometer (Thermo Scientific) and Agilent Bioanalyzer 2100 system. After the library qualified by Qubit and Q-PCR, NovaSeq6000 was used for sequencing and 250 bp paired-end reads were generated. Sequences with ≥97% similarity were assigned to the same OTUs. Shannon index was applied in analyzing complexity of species diversity, calculated with QIIME and displayed with R software.

## Statistical analysis

Using Prism software for graphing and statistical analysis, the experimental results were shown as mean ± SD. Statistical significance of differences between the groups was analyzed with a t-test or one/two-way ANOVA with Šídák's multiple comparisons test. $P < 0.05$ denotes statistical significance.

## Reporting summary

Further information on research design is available in the Nature Portfolio Reporting Summary linked to this article.

## Data availability

The source data supporting the conclusions of this article are provided with this paper. Raw sequences have been deposited in NCBI as Bioproject # PRJNA 981236 and PRJNA 997814. Source data are provided with this paper.

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

## Acknowledgements

This study was supported by National Natural Science Foundation of China (82330017 and 81930015 to W.Z., 82070592 and 82270610 to Y.Y.), National Institutes of Health Grant (R01DK129360 and R01DK112755 to W.Z.), and the Young Elite Scientist Sponsorship Program by CAST (YESS20200034 to Y.Y.).

## Author contributions

Y.L. and C.L.: data acquisition, animal breeding and manuscript drafting; L.S., T.F., W.Y. and Y.Z.: animal breeding, technical and reagents support; M.M.: provided expertise; W.Z. and Y.Y.: experiment design, manuscript revision and funding support. All other authors edited and approved the final manuscript.

## Competing interests

The authors declare no competing interests.
