## [Peer Review File · Nature Communications]

Reduction of specific enterocytes from loss of intestinal LGR4 improves lipid metabolism in miceREVIEWER COMMENTS

Reviewer #1 (Remarks to the Author):

This is an interesting study suggesting that *Lgr4* affects enterocyte differentiation into cells that absorb long-chain fatty acids, being the reason why *Lgr4* KO mice do not gain weight upon HFD to the same extent as WT mice. However these claims are not fully supported by the rather descriptive set of data. I see the following problems/issues

Major:

The link between fat absorption and *Lgr4* is created using scRNAseq. Here the authors identify subsets of enterocytes that express different set of genes. Whether indeed these enterocytes do or do not absorb fat is not clear. No functional data are provided. Organoid experiments could potentially address this. At least some additional validation (e.g. staining of the subsets in the tissue and confirmation of the absence of one of the cell types) should be done. There are many other alternative explanations for epithelial dysfunction upon *Lgr4* depletion, eg. inflammation due to barrier disruption (due to loss of goblet cells), shorter survival time of enterocytes, alterations in microbiota, other defects etc.

From the very beginning upon KO the food intake is different between WT and KO mice, this could have different reasons but may be also the cause of differential weight gains in the WT vs KO mice

The overall proportion of presumably pure fat-absorbing cells is low and the difference between WT and KO are not so obvious, it is therefore not convincing that indeed these alterations are responsible for the phenotypes presented here. What are actually the absolute numbers? The authors say that in KO mice there are more enterocyte progenitors, so I assume there is also an absolute increase in enterocytes? If so, there might be even more fat absorbing cells per crypt in *Lgr4* KO mice..

Differential differentiation in WT vs *Lgr4* KO mice: here, differences in gene expression in progenitor cells are shown as a potential explanation for altered differentiation. Again, organoids should be used to validate that indeed alterations of Wnt signaling cause alterations in enterocyte subsets (e.g. loss of Wnt causes loss of fat-absorbing cells);

Also Organoids from *Lgr4* KO mice should be used to explore whether the effects are direct.

The link to Notch is not clear: why do *Lgr4* KO mice should have altered Notch signaling? Is this R-spo driven?

The authors focus on Lgr4, why not Lgr5. Where is Lgr4 expressed? only in stem cells? In other tissues (stomach, colon), Lgr4 is expressed in almost all cells. This should be explored, as the effect might not be specific to stem cells and instead of lineage commitment the difference could be more direct at the level of enterocytes

Graphic: how do the authors know where the different subsets are present within the crypt?

Discussion: recent insights into Lgr4 signaling (via Rspo and potentially also Rspo independently via osteopontin I think?) should be discussed to deliver a potential mechanism by which Lgr4 KO affects the epithelium. The link to Notch is not clear as well.

Reviewer #2 (Remarks to the Author):

In this study, the authors generated intestinal epithelium-specific Lgr4 knockout mice using Villin-Cre;Lgr4^{flox/flox}, and observed that intestinal Lgr4 deficiency protected mice from HFD-induced obesity and hepatic steatosis. Furthermore, the long-chain fatty acid absorption was decreased, and glucose transport was increased in the intestine. These changes were induced by the decreased enterocytes selective for long-chain fatty acid absorption and the increased carbohydrate-absorptive enterocytes, whose generation may be regulated by Lgr4 via Wnt and Notch signaling pathways. Overall, it is potentially interesting to depict the role of Lgr4 in the lipid and glucose absorption of the intestine, and to reveal the function of Lgr4 in enterocyte differentiation. However, some of the data were not convincing. Also, the mechanism underlying the role of Lgr4 in decreasing the enterocytes selective for long-chain fatty acid absorption and increasing the enterocytes for carbohydrate absorption via Wnt and Notch signaling pathways is unclear.

1. Why the knockout efficiency is so low, only about 50% decrease in both mRNA and protein levels? Did the 50% decrease account for the observed phenotypes?

2. Previous studies revealed that global knockout of Lgr4 impaired the terminal differentiation of Paneth cells and the maintenance of stem cells (Ref. 3-4). In this study, the authors observed the increased proportion of absorptive enterocytes and TA cells, and the decreased proportion of stem cells, goblet cells, Paneth cells, enteroendocrine cells and tuft cells (page 9 line 24-26). The different phenotypes need more detailed investigation. The PAS and LYZ staining have revealed the decreased cell number of goblet and Paneth cells, more staining and quantification should be performed to confirm the changes of other cell types, such as Ki67 staining for TA cells, Olfm4 staining for stem cells, ChgB staining for enteroendocrine cells. ALPI staining should be quantified in Figure S4H, and other staining, and enterocyte ratio should be quantified.

3. The whole changes of enterocytes, but not the subtypes of enterocytes, were revealed by ALPI staining and scRNA-seq and RT-PCR. However, the key conclusion of this study was: “Deficiency of intestinal Lgr4 decreases enterocytes selective for long-chain fatty acid absorption while increasing carbohydrate-absorptive enterocytes” (page 8), which was only observed at the mRNA level (Figure 5). More experiments should be performed to reveal the cell ratio or function changes of enterocyte subtypes. For example, the changes of the proteins participating in long-chain fatty acid and glucose absorption, the immunofluorescent staining and cell ratio quantification of fatty-absorptive enterocytes and carbohydrate-absorptive enterocytes need to be shown; functional experiments, such as glucose absorption in mice or intestinal organoids, to confirm the changes of long-chain fatty acid or glucose absorption should be performed. Without these experiments, the conclusions are mainly based on gene expression.
4. In part 4, the authors made the conclusion that “Intestine-specific knockdown of Lgr4 improves glucose tolerance” according to OGTT (oral glucose administration) and IPGTT (intraperitoneal injection of glucose) (Page 8). However, there were no significant changes in glucose absorption and glucose tolerance between WT and Lgr4iKO mice after oral glucose administration (Figure 4D-E), which don’t support the point “improvement in glucose metabolism and glucose tolerance”. How to explain that the glucose tolerance was improved in IPGTT but showed no changes in OGTT?
5. The authors suggested that: “LGR4 regulates differentiation of intestinal stem cells via Wnt and Notch signaling pathways” according to scRNA-seq (part 6). More functional experiments should be performed to confirm this regulatory mechanism. For example, the ligand or inhibitor treatment of Wnt and Notch signaling in WT and LGR4-KO intestinal organoids.
6. To support the conclusion of decreased lipid absorption and increased glucose absorption in Lgr4iKO, the authors examined the protein levels of FATP4 (Figure 3E) and GLUT2 (Figure 4G). However, the epithelial structures of the small intestine in Lgr4iKO mice were different between Figure 3E and Figure 4G. It seems that the intestinal epithelium was damaged in Lgr4iKO mice in Figure 3E, which may affect the staining conclusion. Therefore, the whole epithelial structures should be showed to confirm whether the integrity of the epithelial structure was changed in Lgr4iKO mice.
7. “leucine-rich repeat G-protein-coupled receptor 4” but not “leucine-rich repeat G-protein-coupled receptor” should be used in title and abstract.
8. Since many genes participate in lipid or glucose transport and metabolism, their functions may be overlapped, such as SLC family. So, several gene expressions can’t represent the whole functional changes, more genes should be examined in Figure 3D, 3H, 4F, et al, so as the staining in Figure 3E and 4G.
9. In Figure 3J, the BODIPY positive cell ratio or fluorescence intensity should be quantified by FACS.
10. It is hard to conclude that: “long-chain fatty acid and carbohydrate-absorptive enterocytes appeared later than long-chain fatty acid-absorptive enterocytes and carbohydrate-absorptive enterocytes” according to trajectory analysis in Figure 5F.
11. There is no description of Figure 3F in the results (part 3).

12. In page 10 line 9-12, the descriptions of figure 6G and figure 6H are reversed. In addition, the protein level of Math1 can't be found in figure 6.
13. The quality of WB was poor in figure 3F, 6F and 6I.
14. The antibody information should be provided in the methods.
15. Statistical analysis should be shown as mean \pm SD, but not as mean \pm SEM. For this reason, some results or significance may be changed.

Reviewer #3 (Remarks to the Author):

In this manuscript by Liang et al., the authors set out to understand the roles of intestinal Lgr4 in regulating lipid absorption. Through a series of genetic and molecular experiments using the Lgr4 epithelial conditional knockout allele (Villin-Lgr4^{fl/fl}), the authors discover that: (1) intestinal specific knockout of Lgr4 results in lower mouse body weight when treated with both normal chow and high fat diets; (2) The Villin-Lgr4^{fl/fl} mouse shows decreased lipid absorption by the gut, and less lipid accumulation in the liver; (3) The glucose transporters (Glut2, Sglt1) positive enterocytes are slightly increased, while the lipid transporters (Cd36, Fatp4)/glucose transporters double positive enterocytes are slightly decreased upon Lgr4 deletion in the gut; (4) The Wnt signal is suppressed, and the Notch signal is activated by Lgr4 deletion.

Overall, the authors present interesting findings that intestinal epithelial deletion of Lgr4 leads to a defect in lipid absorption, thus causing paused (under normal chow) or slower (under high fat diet) mouse bodyweight growth. However, the causal link between bodyweight changes and distinct enterocyte population ratios is a weakness and caveat of this manuscript.

Major points

1. My major criticism of this paper is how a ~10% change in the enterocytes (Figure 5E) could pause mouse bodyweight growth (Figure 1A). In Figure S2, the authors show that Lgr4iKO mice eat less than their littermate control mice, indicating that the mice may be ill. It would be nice to show these mice's body condition score (BCS).
2. It has been reported that Lgr4 is required for Paneth maintenance (PMID: 23393138), and the authors also show Lgr4iKO causes depletion in Paneth and Tuft cell numbers (Figure 6A). It is demonstrated that

Paneth cell disruption could lead to gut defects, such as enterocolitis. Are the whole-body phenotypes in Lgr4iKO mice possibly due to Paneth or Tuft cell defects?

3. In Figure 3D-E, the authors show that lipid transport genes (Fatp4, Cd36) are downregulated in Lgr4iKO mouse intestine. From Figure 3E, the reduction of gene expression happens in the tip of the microvillus. Hans Clevers' group showed that Lgr4 expression is restricted to the crypts (PMID: 21727895). How does LGR4 in crypts change gene expression at the top of the microvillus? Also, Beyaz et al. showed that Fatp4 (Slc27a4) is expressed in the Lgr5+ stem cells (PMID: 26935695). It would be helpful to check Fatp4 expression in Lgr4iKO crypt cells. In Figure 3G-H, the authors use MODE-K cells as an in vitro model to study the regulation of Lgr4 on lipid metabolism cell-autonomously. As discussed above, Lgr4 is not expressed in the lipid absorptive enterocytes; thus, the MODE-K model does not recapitulate the in vivo biology.

4. In Figure 4D-E, the authors propose that intestinal glucose absorption was increased in Lgr4iKO mice. More evidence needs to be provided to substantiate this statement. Moreover, if the glucose absorption is increased in Lgr4iKO mice, does supplementation with glucose water restore/rescue body growth?

5. In Figure 5A-B, the scRNAseq sample size is labeled as N=3, does this mean there are 3 biological replicates for each condition, or the total number of mice for WT and Lgr4iKO conditions is 3? In Figure 5E, 6A, the error bar is missing, and the broken Y-axis makes the difference look bigger than it is (i.e., the data graphics distorts how big the difference is, which can be misleading at first glance).

6. It is established that Paneth cells support ISCs via the expression of Notch ligands. In the absence of Paneth cells in Lgr4iKO, what is the mechanism for enhanced Notch signaling?

7. The authors mentioned that Lgr4flox/flox mice were purchased from Jackson Lab; please provide the stock number, as this strain could not be found on JAX's website.

Thank you for your time and effort to review our manuscript “Deficiency of intestinal leucine-rich repeat G-protein-coupled receptor improves lipid metabolism by reducing the proportion of enterocytes selective for long-chain fatty acid absorption” (ID: NCOMMS-23-03398). We have found the reviewers’ comments informative and constructive. The point-by-point response follows:

Reviewer 1:

This is an interesting study suggesting that *Lgr4* affects enterocyte differentiation into cells that absorb long-chain fatty acids, being the reason why *Lgr4* KO mice do not gain weight upon HFD to the same extent as WT mice. However, these claims are not fully supported by the rather descriptive set of data. I see the following problems/issues

Major:

1. The link between fat absorption and *Lgr4* is created using scRNA seq. Here the authors identify subsets of enterocytes that express different set of genes. Whether indeed these enterocytes do or do not absorb fat is not clear. No functional data are provided. Organoid experiments could potentially address this. At least some additional validation (e.g. staining of the subsets in the tissue and confirmation of the absence of one of the cell types) should be done. There are many other alternative explanations for epithelial dysfunction upon *Lgr4* depletion, eg. inflammation due to barrier disruption (due to loss of goblet cells), shorter survival time of enterocytes, alterations in microbiota, other defects etc.

Reply: Per suggestion, we have performed the organoid experiments and confirmed that Lgr4 deficiency reduces the lipid absorption evidenced by down-regulation of Fabp1 (Attached figure 1). Together with following observations, our study indicates that deficiency of intestinal Lgr4 decreases enterocytes selective for long-chain fatty acid absorption while increasing carbohydrate-absorptive enterocytes. First, key intestinal lipid transporters including FATP4 (Fig. 3e) (Immunohistochemical staining, Western blotting and q-RT-PCR) were decreased, whereas carbohydrate transporters such as GLUT2 (Fig. 4g) increased. Second, Oil red O staining indicates a significant reduction in lipid content in Lgr4-deficient enterocytes (Figure 3c). Third, Lgr4 deficiency reduces lipid absorption in cultured enterocytes (Figure 3j).

Attached figure 1. Morphology and mRNA levels of lipid absorption markers in organoid. β -actin was used as a reference gene. * $P < 0.05$ vs WT. $n = 3$.

We agree with the reviewer that there may exist other alternative explanations such as intestinal barrier disruption, inflammation, alteration in microbiota as well as other defects (Fig. S6 a-d and Fig. S7). However, we believe that these alternative mechanisms are unlikely contributing to the phenotypes because these alternations typically lead to either the simultaneous increase or decrease in the absorption of lipid and carbohydrate. Further, the observation that *Lgr4* deficiency suppresses the apoptosis indicates that the survival time of enterocytes is increased instead of decrease (Fig. S6e)

2. From the very beginning upon KO the food intake is different between WT and KO mice, this could have different reasons but may be also the cause of differential weight gains in the WT vs KO mice.

Reply: Decrement in food intake was only observed in NCD-fed $Lgr4^{iKO}$ mice. In mice fed HFD, the difference in food intake between WT and $Lgr4^{iKO}$ mice was not significant (Fig. S2a), but the body weight was still significantly reduced (Fig. 1b). This observation suggests that reduced food intake was a contributor, but not the only cause of weight loss in $Lgr4^{iKO}$ mice.

3. The overall proportion of presumably pure fat-absorbing cells is low and the difference between WT and KO are not so obvious, it is therefore not convincing that indeed these alterations are responsible for the phenotypes presented here. What are actually the absolute numbers?

The authors say that in KO mice there are more enterocyte progenitors, so I assume

there is also an absolute increase in enterocytes? If so, there might be even more fat absorbing cells per crypt in Lgr4 KO mice.

Reply: We agree with the reviewer that the overall proportion of pure fat-absorbing cells is low. Given the fact that enterocytes are numerous, small change in the proportion of long-chain fatty acid -absorptive enterocytes may result in a large enough difference in the actual quantity to present phenotypes. These assumption is supported by the observation that absorption of lipid was significantly reduced upon oral gavage of lipid (Figure 3b). Although we are unable to isolate these pure fat-absorbing cells, our immunohistochemical staining on FATP4 (Figure 3e) indicates that these fat-absorbing cells were decreased in Lgr4 deficient mice.

4. Differential differentiation in WT vs Lgr4 KO mice: here, differences in gene expression in progenitor cells are shown as a potential explanation for altered differentiation. Again, organoids should be used to validate that indeed alterations of Wnt signaling cause alterations in enterocyte subsets (e.g. loss of Wnt causes loss of fat-absorbing cells); Also Organoids from lgr4 KO mice should be used to explore whether the effects are direct.

The link to Notch is not clear: why do Lgr4 KO mice should have altered Notch signaling? Is this R-spo driven?

Reply: Since culture medium for organoid contains R-spondin and other activators of Wnt signaling, it is impossible to address this issue using the organoid culture. Instead, we performed additional experiments using cultured MODE-K enterocytes. As shown in Fig. 6j, activation of Notch signaling by VPA significantly suppresses the Wnt signaling evidenced by the decrement of its relevant target genes. This observation suggests a link between Wnt and Notch signaling. In addition, inhibition of Wnt signaling substantially decreases expression of lipid uptake genes fatp4 in MODE-K cells (Fig. 6k). This observation suggests that inhibition of Wnt signaling may decrease lipid absorption.

5. The authors focus on Lgr4, why not Lgr5. Where is Lgr4 expressed? only in stem cells? In other tissues (stomach, colon), Lgr4 is expressed in almost all cells. This should be explored, as the effect might not be specific to stem cells and instead of lineage commitment the difference could be more direct at the level of enterocytes.

Reply: The role of LGR5 in intestinal development has been extensively studied. Whether intestinal LGR4 impacts intestinal development and energy homeostasis remains unknown. Based on the results of single-cell sequencing, LGR4 is expressed at the highest level in stem cells, followed by goblet cells and enteroendocrine cells (Attached fig. 2). Since levels of Lgr4 is highest in stem cells, we focused our study on the stem cells.

Attached figure 2. Violin plots for expression level of Lgr4 in scRNA-seq analysis.

6. Graphic: how do the authors know where the different subsets are present within the crypt?

Reply: Per suggestion, we have revised the graphic.

7. Discussion: recent insights into Lgr4 signaling (via Rspo and potentially also Rspo independently via osteopontin I think?) should be discussed to deliver a potential mechanism by which Lgr4 KO affects the epithelium. The link to Notch is not clear as well.

Reply: We have addressed recent insights into LGR4 signaling in lines 6-12, page 9 in the discussion section.

Reviewer #2 (Remarks to the Author):

In this study, the authors generated intestinal epithelium-specific Lgr4 knockout mice using Villin-Cre; Lgr4^{flox/flox}, and observed that intestinal Lgr4 deficiency protected mice from HFD-induced obesity and hepatic steatosis. Furthermore, the long-chain fatty acid absorption was decreased, and glucose transport was increased in the intestine. These changes were induced by the decreased enterocytes selective for long-chain fatty acid absorption and the increased carbohydrate-absorptive enterocytes, whose generation may be regulated by Lgr4 via Wnt and Notch signaling pathways. Overall, it is potentially interesting to depict the role of Lgr4 in the lipid and glucose absorption of the intestine, and to reveal the function of Lgr4 in enterocyte differentiation. However, some of the data were not convincing. Also, the mechanism underlying the role of Lgr4 in decreasing the enterocytes selective for long-chain fatty acid absorption and increasing the enterocytes for carbohydrate absorption via Wnt and Notch signaling pathways is unclear.

1. Why the knockout efficiency is so low, only about 50% decrease in both mRNA and protein levels? Did the 50% decrease account for the observed phenotypes?

Reply: Although Lgr4 is mainly expressed in intestinal epithelial cells, other cells in the gut may also express Lgr4. This may explain why LGR4 protein was almost absent, while its mRNA levels were decreased only by about 60%. All the data indicates that deficiency of LGR4 in enterocytes is sufficient for the phenotypes.

2. Previous studies revealed that global knockout of *Lgr4* impaired the terminal differentiation of Paneth cells and the maintenance of stem cells (Ref. 3-4). In this study, the authors observed the increased proportion of absorptive enterocytes and TA cells, and the decreased proportion of stem cells, goblet cells, Paneth cells, enteroendocrine cells and tuft cells (page 9 line 24-26). The different phenotypes need more detailed investigation. The PAS and LYZ staining have revealed the decreased cell number of goblet and Paneth cells, more staining and quantification should be performed to confirm the changes of other cell types, such as Ki67 staining for TA cells, *Olfm4* staining for stem cells, *ChgB* staining for enteroendocrine cells. ALPI staining should be quantified in Figure S4H, and other staining, and enterocyte ratio should be quantified.

Reply: Per suggestion, Ki67 staining for TA cells, Olfm4 staining for stem cells and quantification of ALPI are now included in Fig S4f, Fig S4c and Fig S4j. Enteroendocrine cells were stained by Grimelius's silver nitrate method and the number of enteroendocrine cells per villus-crypt unit was included in Fig S5i.

3. The whole changes of enterocytes, but not the subtypes of enterocytes, were revealed by ALPI staining and scRNA-seq and RT-PCR. However, the key conclusion of this study was: "Deficiency of intestinal *Lgr4* decreases enterocytes selective for long-chain fatty acid absorption while increasing carbohydrate-absorptive enterocytes" (page 8), which was only observed at the mRNA level (Figure 5). More experiments should be performed to reveal the cell ratio or function changes of enterocyte subtypes. For example, the changes of the proteins participating in long-chain fatty acid and glucose absorption, the immunofluorescent staining and cell ratio quantification of fatty-absorptive enterocytes and carbohydrate-absorptive enterocytes need to be shown.

Functional experiments, such as glucose absorption in mice or intestinal organoids, to confirm the changes of long-chain fatty acid or glucose absorption should be performed. Without these experiments, the conclusions are mainly based on gene expression.

Reply: Per suggestion, quantification of FATP4 and GLUT2 IHC staining is now included in Fig 3e and Fig 4g.

We agree with the reviewer that functional experiments in mice and organoids will provide solid evidence supporting our conclusion. Thus, additional experiment using organoid was performed. As shown in Attached figure 1, Lgr4 deficiency significantly inhibits the expression of lipid uptake gene Fabp1. In addition, following observations in mice and cultured enterocytes further support our conclusion. First, key intestinal lipid transporters including FATP4 (Fig. 3e) (Immunohistochemical staining, Western blotting and q-RT-PCR) were decreased, whereas carbohydrate transporters such as GLUT2 (Fig. 4g) increased. Second, Oil red O staining indicates a significant reduction in lipid content in Lgr4-deficient enterocytes (Figure 3c). Third, Lgr4 deficiency

reduces lipid absorption in cultured enterocytes (Figure 3j).

Attached figure 1. Morphology and mRNA levels of lipid absorption markers in organoid. β -actin was used as a reference gene. * $P < 0.05$ vs WT. $n = 3$.

4. In part 4, the authors made the conclusion that “Intestine-specific knockdown of Lgr4 improves glucose tolerance” according to OGTT (oral glucose administration) and IPGTT (intraperitoneal injection of glucose) (Page 8). However, there were no significant changes in glucose absorption and glucose tolerance between WT and Lgr4iKO mice after oral glucose administration (Figure 4D-E), which don’t support the point “improvement in glucose metabolism and glucose tolerance”. How to explain that the glucose tolerance was improved in IPGTT but showed no changes in OGTT?

Reply: Lgr4^{iKO} mice showed a decrease of glucose AUC during IPGTT (Fig 4a and b), but not OGTT (Fig 4d and e). This result suggests that deficiency of intestinal Lgr4 increases glucose tolerance and glucose absorption. In OGTT, the improvement in glucose tolerance and concurrent increment in glucose absorption led to no change in OGTT.

5. The authors suggested that: “LGR4 regulates differentiation of intestinal stem cells via Wnt and Notch signaling pathways” according to scRNA-seq (part 6). More functional experiments should be performed to confirm this regulatory mechanism. For example, the ligand or inhibitor treatment of Wnt and Notch signaling in WT and LGR4-KO intestinal organoids.

Reply: Per suggestion, we performed additional experiments using cultured MODE-K enterocytes. As shown in Fig 6j, activation of Notch signaling by VPA significantly

suppresses the Wnt signaling evidenced by the decrement of its relevant target genes. This observation suggests a link between Wnt and Notch signaling. Since culture medium for organoid contains R-spondin and other activators of Wnt signaling, it is impossible to test this concept in the cultured organoid.

6. To support the conclusion of decreased lipid absorption and increased glucose absorption in Lgr4iKO, the authors examined the protein levels of FATP4 (Figure 3E) and GLUT2 (Figure 4G). However, the epithelial structures of the small intestine in Lgr4iKO mice were different between Figure 3E and Figure 4G. It seems that the intestinal epithelium was damaged in Lgr4iKO mice in Figure 3E, which may affect the staining conclusion. Therefore, the whole epithelial structures should be showed to confirm whether the integrity of the epithelial structure was changed in Lgr4iKO mice.

Reply: We have re-performed and re-analyzed FATP4 staining (Fig 3e). The whole epithelial structures were now showed in Figure 3.

7. “Leucine-rich repeat G-protein-coupled receptor 4” but not “leucine-rich repeat G-protein-coupled receptor” should be used in title and abstract.

Reply: We have revised the statement accordingly in the title and abstract.

8. Since many genes participate in lipid or glucose transport and metabolism, their functions may be overlapped, such as SLC family. So, several gene expressions can't represent the whole functional changes, more genes should be examined in Figure 3D, 3H, 4F, et al, so as the staining in Figure 3E and 4G.

Reply: Per suggestion, more genes were analyzed and now included in Fig 3d and 3h. For lipid transporter, Fabp2 was analyzed and showed no change (Figure 3D). For the glucose transporters of SLC family such as Glut7, Glut8 and Glut12, their CT values were about 29, indicating a low expression level. Thus, we do not believe that these genes play a significant role.

9. In Figure 3J, the BODIPY positive cell ratio or fluorescence intensity should be quantified by FACS.

Reply: The FACS analysis data is now included in Fig 3j.

10. It is hard to conclude that: “long-chain fatty acid and carbohydrate-absorptive enterocytes appeared later than long-chain fatty acid-absorptive enterocytes and carbohydrate-absorptive enterocytes” according to trajectory analysis in Figure 5F.

Reply: We agree with the reviewer, and omitted this statement accordingly.

11. There is no description of Figure 3F in the results (part 3).

Reply: The description of Fig 3f is now included in lines 7-9, page 5.

12. In page 10 line 9-12, the descriptions of figure 6G and figure 6H are reversed. In addition, the protein level of Math1 can't be found in figure 6.

Reply: We have corrected this mistake in the manuscript.

13. The quality of WB was poor in figure 3F, 6F and 6I.

Reply: Per suggestion, we have re-performed the WB of Fig 3f, 6f and 6i.

14. The antibody information should be provided in the methods.

Reply: We have listed all the antibodies and relevant information in lines 33-35, page 9 and lines 1-6, page 10 in the method section.

15. Statistical analysis should be shown as mean \pm SD, but not as mean \pm SEM. For this reason, some results or significance may be changed.

Reply: We have changed the statistical analysis from mean \pm SEM to mean \pm SD accordingly.

Reviewer #3 (Remarks to the Author):

In this manuscript by Liang et al., the authors set out to understand the roles of intestinal Lgr4 in regulating lipid absorption. Through a series of genetic and molecular experiments using the Lgr4 epithelial conditional knockout allele (Villin-Lgr4fl/fl), the authors discover that: (1) intestinal specific knockout of Lgr4 results in lower mouse body weight when treated with both normal chow and high fat diets; (2) The Villin-Lgr4fl/fl mouse shows decreased lipid absorption by the gut, and less lipid accumulation in the liver; (3) The glucose transporters (Glut2, Sglt1) positive enterocytes are slightly increased, while the lipid transporters (Cd36, Fatp4)/glucose transporters double positive enterocytes are slightly decreased upon Lgr4 deletion in the gut; (4) The Wnt signal is suppressed, and the Notch signal is activated by Lgr4 deletion.

Overall, the authors present interesting findings that intestinal epithelial deletion of Lgr4 leads to a defect in lipid absorption, thus causing paused (under normal chow) or slower (under high fat diet) mouse bodyweight growth. However, the causal link between bodyweight changes and distinct enterocyte population ratios is a weakness and caveat of this manuscript.

Major points

1. My major criticism of this paper is how a ~10% change in the enterocytes (Figure 5E) could pause mouse bodyweight growth (Figure 1A). In Figure S2, the authors show that Lgr4iKO mice eat less than their littermate control mice, indicating that the mice may be ill. It would be nice to show these mice's body condition score (BCS).

Reply: Per suggestions, the body condition score (BCS) was analyzed and included in Fig 1c. The results showed that Lgr4^{iKO} mice were healthy. Food intake of NCD-fed Lgr4^{iKO} mice decreased. However, the difference in food intake between WT and Lgr4^{iKO} mice fed HFD was not significant (Fig. S2a), but the body weight was still significantly reduced (Fig. 1b), suggesting that reduced food intake was a contributor, but not the only cause, of weight loss in Lgr4^{iKO} mice.

2. It has been reported that Lgr4 is required for Paneth maintenance (PMID: 23393138), and the authors also show Lgr4iKO causes depletion in Paneth and Tuft cell numbers (Figure 6A). It is demonstrated that Paneth cell disruption could lead to gut defects, such as enterocolitis. Are the whole-body phenotypes in Lgr4iKO mice possibly due to Paneth or Tuft cell defects?

Reply: Consistent with the decrement in the number of Paneth cells and Tuft cells, the barrier disruption and alteration of the microbiota were observed (Fig. S6 and Fig. S7). However, we believe that these alterations are unlikely contributing to the phenotypes because these changes typically lead to either the simultaneous increase or decrease in the absorption of lipid and carbohydrate. In contrast, we observed a reduction in lipid absorption with concurrent increase in glucose absorption. In addition, mice appear to be healthy based on the body condition score analysis (Fig 1c).

3. In Figure 3D-E, the authors show that lipid transport genes (Fatp4, Cd36) are downregulated in Lgr4iKO mouse intestine. From Figure 3E, the reduction of gene expression happens in the tip of the microvillus. Hans Clevers' group showed that Lgr4 expression is restricted to the crypts (PMID: 21727895). How does LGR4 in crypts change gene expression at the top of the microvillus? Also, Beyaz et al. showed that Fatp4 (Slc27a4) is expressed in the Lgr5+ stem cells (PMID: 26935695). It would be helpful to check Fatp4 expression in Lgr4iKO crypt cells. In Figure 3G-H, the authors use MODE-K cells as an in vitro model to study the regulation of Lgr4 on lipid metabolism cell-autonomously. As discussed above, Lgr4 is not expressed in the lipid absorptive enterocytes; thus, the MODE-K model does not recapitulate the in vivo biology.

Reply: Our results of single-cell sequencing indicate that *LGR4* is expressed in absorptive enterocytes, not restricted to the crypts (Attached figure 2). In addition, our additional experiments using organoid provide further evidence supporting our conclusion that *Lgr4* deficiency significantly inhibits the expression lipid uptake gene *Fabp1*. (Attached figure 1).

Attached figure 1. Morphology and mRNA levels of lipid absorption markers in organoid. β -actin was used as a reference gene. * $P < 0.05$ vs WT. $n = 3$.

Attached figure 2. Violin plots for expression level of *Lgr4* in scRNA-seq analysis.

4. In Figure 4D-E, the authors propose that intestinal glucose absorption was increased in *Lgr4*^{iKO} mice. More evidence needs to be provided to substantiate this statement. Moreover, if the glucose absorption is increased in *Lgr4*^{iKO} mice, does supplementation with glucose water restore/rescue body growth?

Reply: We agree with the reviewer that more evidence is needed to support our conclusion on intestinal glucose absorption. Indeed, our data showing that the difference in body weight in mice fed HFD between WT and *Lgr4*^{iKO} mice was less significant than those animals fed NCD (Fig. 1a, 1b) indicates that increased energy supply could at least partially reverse the weight loss in *Lgr4*^{iKO} mice. We did not

perform the rescue experiment using supplementation with glucose water because this approach may complicate the intestinal microecology (Kawano Y et al, Cell 2022).

5. In Figure 5A-B, the scRNAseq sample size is labeled as N=3, does this mean there are 3 biological replicates for each condition, or the total number of mice for WT and Lgr4iKO conditions is 3? In Figure 5E, 6A, the error bar is missing, and the broken Y-axis makes the difference look bigger than it is (i.e., the data graphics distorts how big the difference is, which can be misleading at first glance).

Reply: Each group contained 3 mice.

The Y-axis is broken to focus the presentation on pure fat-absorbing or carbohydrate-absorbing cells.

6. It is established that Paneth cells support ISCs via the expression of Notch ligands. In the absence of Paneth cells in Lgr4iKO, what is the mechanism for enhanced Notch signaling?

Reply: It has been reported that in mouse models displaying Paneth cell depletion, enteroendocrine and tuft cells can serve as an alternative source for Notch signals, thus supporting ISC maintenance (van Es JH et al, Proc Natl Acad Sci U S A 2019). This shows that the adaptive flexibility of the intestine in maintaining normal tissue homeostasis and Paneth cells are not the only source of notch. However, the mechanism for enhanced Notch signaling in the absence of Paneth cells still remains unknown.

7. The authors mentioned that Lgr4flox/flox mice were purchased from Jackson Lab; please provide the stock number, as this strain could not be found on JAX's website.

Reply: Sorry for the mistake. The transgene was established by our lab and reported in our previous publications (Ziru Li et al, Am J Physiol Gastrointest Liver Physiol 2019). The relevant information is now included in line 16, page 10 in the methods section.

REVIEWER COMMENTS

Reviewer #1 (Remarks to the Author):

The paper is improved now and most point have been sufficiently adressed.

Particularly, Organoids were now used to explore the direct effect on epithelial cells - to me this assay is essential to exclude unspecific effects, driven by other alterations on microbiota/immune system etc. It is therefore extremely surprising that the data are included to the rebuttal but not included into the manuscript. Why is this?

In this context I believe it is important to expand the data obtained in organoids: the authors claim a highly specific effect on enterocytes for long-chain fatty acids - it will be important to show this, for now only expression of FapB1 is seen, while the effects on other genes and pathways that are (or are not) affected in vivo) are not displayed.

Is there also a way to measure lipid absortion in organoids?

Reviewer #2 (Remarks to the Author):

The authors revised the manuscript and most of the concerns have been addressed. Still, one small concern remains: the author performed IPGTT and OGTT and concluded that that “deficiency of intestinal Lgr4 increases glucose tolerance and glucose absorption” (Rebuttal reply 4; Revised manuscript, page 5, line 155-156), to explain the glucose tolerance in OGTT. However, the blood glucose showed no significant difference between WT and Lgr4iKO mice in 0-15 min (the main absorption period) after oral glucose administration (Fig 4d-e), which suggest that the glucose absorption may not be changed in Lgr4iKO mice. Although glucose transporters were increased in mRNA and protein level (Fig 4f-g), other glucose absorption experiments should be performed to confirm the glucose functional enhancement.

Reviewer #3 (Remarks to the Author):

While the authors addressed some of our concerns, we do not have additional (unanswered) questions for the authors. We are not convinced that the authors have satisfactorily addressed previous concerns; thus, dampening our enthusiasm for the publication of this manuscript in Nature Comm. Below are some specific points that need to be addressed:

1) The authors have not completely addressed the lack of functional studies, which would provide conclusive evidence that Lgr4 KO results in decreased fat absorption, and that would rule out the contribution of additional factors such as intestinal barrier disruption, inflammation, microbiota changes etc. For instance, experiments in Lgr4 KO organoid cultures would rule out some of these alternate factors, but they only show downregulation of Fabp1 (is there possible upregulation of other transporters to compensate?), but it would be nice to actually show decreased lipid content in these organoids. Just down-regulation of Fabp1 by itself does not seem convincing.

2) Evidence to support decrease in "enterocytes selective for absorption of long-chain fatty acid". This is limited to 1) sc-RNA-Seq, which need further validation, 2) mRNA and/or protein quantification of CD36, Fabp4 and fatp2 on the bulk intestine that does not provide evidence for their claim that among the heterogeneity of the enterocytes, a specific subset of fat absorbing cells is increased, and 3) IHC of FATP4 where the quantification is limited to % positive area. A proper quantification of these cells that they define in the sc-RNA-Seq as "enterocytes selective for absorption of long-chain fatty acid" is needed using specific markers that can be represented as proportion and absolute number of cells in the intestine. This should also be validated in the organoids.

3) The mechanism by which Lgr4 KO cells have altered Notch signaling is still not clear.

Thank you for your time and effort to review our manuscript “Deficiency of intestinal leucine-rich repeat G-protein-coupled receptor 4 improves lipid metabolism by reducing the proportion of enterocytes selective for long-chain fatty acid absorption” (ID: NCOMMS-23-03398). We have found the reviewers’ comments informative and constructive. The point-by-point response follows:

Reviewer #1 (Remarks to the Author):

The paper is improved now and most point have been sufficiently addressed.

Particularly, Organoids were now used to explore the direct effect on epithelial cells - to me this assay is essential to exclude unspecific effects, driven by other alterations on microbiota/immune system etc. It is therefore extremely surprising that the data are included to the rebuttal but not included into the manuscript. Why is this?

In this context I believe it is important to expand the data obtained in organoids: the authors claim a highly specific effect on enterocytes for long-chain fatty acids - it will be important to show this, for now only expression of FapB1 is seen, while the effects on other genes and pathways that are (or are not) affected in vivo) are not displayed. Is there also a way to measure lipid absorption in organoids?

Reply: Per suggestion, we measured lipid absorption in organoids using BodipyTM 500/510 C1, C12 and found that the deficiency of Lgr4 impaired the ability of fatty acid absorption in organoids (Fig S4c). We quantified the lipid absorption related genes of organoids using q-PCR and found a significant decrease of Fabp1 (Fig S4b). In summary, we have included the effect of Lgr4 deficiency on the growth process (Fig S4a), the gene changes (Fig S4b) and fatty acid absorption (Fig S4c) of the organoid.

Reviewer #2 (Remarks to the Author):

The authors revised the manuscript and most of the concerns have been addressed. Still, one small concern remains: the author performed IPGTT and OGTT and concluded that that “deficiency of intestinal Lgr4 increases glucose tolerance and glucose absorption” (Rebuttal reply 4; Revised manuscript, page 5, line 155-156), to explain the glucose tolerance in OGTT. However, the blood glucose showed no significant difference between WT and Lgr4iKO mice in 0-15 min (the main absorption period) after oral glucose administration (Fig 4d-e), which suggest that the glucose absorption may not be changed in Lgr4iKO mice. Although glucose transporters were increased in mRNA and protein level (Fig 4f-g), other glucose absorption experiments should be performed to confirm the glucose functional enhancement.

Reply: Phase analysis can be very useful in IPGTT because glucose is quickly and almost equally absorbed into blood and induces insulin secretion and blood glucose level up (glucose directly absorbed) and down (after insulin functions) in the first 15 minutes. But in OGTT, phase analysis may be of less significance because the glucose level is complicatedly affected by glucose intake, insulin secretion, insulin resistance and other factors. We found that the glucose tolerance of $Lgr4^{iKO}$ mice was improved, and more glucose intake might cause a more rapid and acute insulin secretion and response, leading to rapid clearance of circulating glucose. This may explain that $Lgr4^{iKO}$ mice would not show a higher blood glucose level at the time point of 15 minutes despite the level of glucose transporters were increased.

Reviewer #3 (Remarks to the Author):

While the authors addressed some of our concerns, we do not have additional (unanswered) questions for the authors. We are not convinced that the authors have satisfactorily addressed previous concerns; thus, dampening our enthusiasm for the publication of this manuscript in Nature Comm. Below are some specific points that need to be addressed:

1) The authors have not completely addressed the lack of functional studies, which would provide conclusive evidence that $Lgr4$ KO results in decreased fat absorption, and that would rule out the contribution of additional factors such as intestinal barrier disruption, inflammation, microbiota changes etc. For instance, experiments in $Lgr4$ KO organoid cultures would rule out some of these alternate factors, but they only show downregulation of *Fabp1* (is there possible upregulation of other transporters to compensate?), but it would be nice to actually show decreased lipid content in these organoids. Just down-regulation of *Fabp1* by itself does not seem convincing.

Reply: Additional experiments have been performed to measure the lipid absorption and other genes related to lipid absorption. These results are now included in Fig S4b,c.

2) Evidence to support decrease in "enterocytes selective for absorption of long-chain fatty acid". This is limited to 1) sc-RNA-Seq, which need further validation, 2) mRNA and/or protein quantification of CD36, *Fabp4* and *fatp2* on the bulk intestine that does not provide evidence for their claim that among the heterogeneity of the enterocytes, a specific subset of fat absorbing cells is increased, and 3) IHC of *FATP4* where the quantification is limited to % positive area. A proper quantification of these cells that they define in the sc-RNA-Seq as "enterocytes selective for absorption of long-chain fatty acid" is needed using specific markers that can be represented as proportion and absolute number of cells in the intestine. This should also be validated in the organoids.

Reply: The ability of glucose and lipid intake in intestine is mainly mediated by the transporters in epithelia cells. Therefore, we defined the subgroups of enterocytes using the expression of these absorption-related genes. In addition, we confirmed our

observations by measurement of lipid absorption and related genes in organoid.

3) The mechanism by which Lgr4 KO cells have altered Notch signaling is still not clear.

Reply: Per suggestion, we further explored the relationship between Lgr4 and Notch signaling in our research. In MODE-K cells, we quantified the mRNA levels of genes in Notch signaling, and found that levels of Psen1, Aph1a and Nedd4 were significantly changed (Fig S9a). We then further confirmed the increment of PSEN1 mRNA (Fig S9b) and protein levels (Fig S9c) in IEC6 cells. PSEN1 is an active component of γ -secretase, which is responsible for the cleavage of Notch and next-step function of NICD. This observations suggest that deficiency of Lgr4 enhances the expression of Psen1 and thus stimulating the function of Notch signaling. The canonical pathway of LGR4 function is mediated by the nuclear translocation of β -catenin and then the transcriptional regulation of TCF/LEF transcription factor family. Using UCSC Genome Browser Home and JASPAR database, we predicted that TCF7L2 may bind to the promoter region of Psen1. Therefore, we constructed pcDNA3.4-Tcf7l2 and pGL3-Psen1 promoter plasmids and cotransfected them with pRL-TK into 293T cells. We found that Psen1 relative luciferase activity was significantly reduced in condition of Tcf7l2 overexpression (Fig S9d). Together, these results suggest that deficiency of LGR4 may activate Notch signaling via β -catenin and TCF7L2 mediated transcriptional regulation of Psen1.

REVIEWER COMMENTS

Reviewer #1 (Remarks to the Author):

To simplify the communication I have reproduced the main point from my previous request, the response and added a new comment:

Particularly, Organoids were now used to explore the direct effect on epithelial cells - to me this assay is essential to exclude unspecific effects, driven by other alterations on microbiota/immune system etc. It is therefore extremely surprising that the data are included to the rebuttal but not included into the manuscript. Why is this?

In this context I believe it is important to expand the data obtained in organoids: the authors claim a highly specific effect on enterocytes for long-chain fatty acids - it will be important to show this, for now only expression of FapB1 is seen, while the effects on other genes and pathways that are (or are not) affected *in vivo* are not displayed. Is there also a way to measure lipid absorption in organoids?

Reply: Per suggestion, we measured lipid absorption in organoids using BodipyTM 500/510 C1, C12 and found that the deficiency of Lgr4 impaired the ability of fatty acid absorption in organoids (Fig S4c). We quantified the lipid absorption related genes of organoids using q-PCR and found a significant decrease of Fapb1 (Fig S4b). In summary, we have included the effect of Lgr4 deficiency on the growth process (Fig S4a), the gene changes (Fig S4b) and fatty acid absorption (Fig S4c) of the organoid.

New:

The BodipyTM data - as currently presented in the figure - are not really helpful, there is one picture, no quantification, no information on the number of replicates etc is provided. In the present form the data do not address the request and are not convincing. Please provide data from several replicates and quantify the signal!

As I have mentioned, there is only one gene FabBp1 presented in the figure to be regulated in Lgr4Ko in organoids. While it is really nice to see the organoid data in the ms now, the request remains to provide a more comprehensive characterization of organoids, what other key genes are or are not expressed in Lgr4 dependent manner?

Reviewer #2 (Remarks to the Author):

No more questions

Reviewer #3 (Remarks to the Author):

Although the authors have addressed most of my concerns, there still remain concerns that have not been satisfactorily addressed and need some clarification before publication, as listed below.

1) Concerns regarding the claim of a decrease in enterocytes selective for absorption of long-chain fatty acid have not been satisfied. As mentioned, some validation studies regarding the proportion and absolute number of LCFA absorptive cells in the intestine would benefit the claim.

2) The link between Lgr4 and Notch signaling is still not clear. While more mechanistic studies in cell lines have shed some light, it is crucial to carry out functional experiments, including loss-of-function validations, in more relevant organoid or in-vivo systems to establish a clear connection between Lgr4 deficiency and activated Notch signaling or clarify the gaps in knowledge regarding this in the text.

Reviewer #1 (Remarks to the Author):

New:

The BodipyTM data - as currently presented in the figure are not really helpful, there is one picture, no quantification, no information on the number of replicates etc is provided. In the present form the data do not address the request and are not convincing. Please provide data from several replicates and quantify the signal!

Reply: We are really sorry for this negligence. We have now included the number of experiments and quantification data in Fig. S4c. The difference is statistically significant.

As I have mentioned, there is only one gene FabBp1 presented in the figure to be regulated in Lgr4Ko in organoids. While it is really nice to see the organoid data in the ms now, the request remains to provide a more comprehensive characterization of organoids, what other key genes are or are not expressed in Lgr4 dependent manner?

Reply: Per suggestion, we have repeated the experiments and detected more genes related to lipid absorption such as Fabp1, Fatp4, Scarb1 and Cav1. We found a significant reduction in Fabp1 and Fatp4, and a tendency of reduction in Scarb1, whereas Cav1 remained unaltered. Together with the Bodipy functional experiments, these observations provide evidence supporting our hypothesis. To further confirm our results in single cell RNA-seq, we also measured the mRNA levels of different cell markers, and found that those of stem cell, Paneth cell and Tuft cell were significantly decreased (Fig. S6l). These findings further support our hypothesis.

Reviewer #2 (Remarks to the Author):

No more questions

Reviewer #3 (Remarks to the Author):

Although the authors have addressed most of my concerns, there still remain concerns that have not been satisfactorily addressed and need some clarification before publication, as listed below.

1) Concerns regarding the claim of a decrease in enterocytes selective for absorption of long-chain fatty acid have not been satisfied. As mentioned, some validation studies regarding the proportion and absolute number of LCFA absorptive cells in the intestine would benefit the claim.

Reply: Our data on organoid BODIPY absorption (Fig. S4c) and FATP4 IHC (Fig. 3e) experiments showed a heterogeneity in enterocytes.

In order to provide some quantitative data, we circled every single cell in organoids and quantified its mean fluorescent intensity. As shown in figure S4c, the number of cells absorbing more lipid defined by the mean fluorescence intensity of BODIPY was decreased in organoids derived from Lgr4 deficient mice.

In addition, we quantified the signal intensity of FATP4 immunoreactivity in FATP4 IHC figures to define the different levels of FATP4 as low, medium and high. As shown in figure 3e, area of medium

and high FATP4 expression was decreased in *Lgr4^{iKO}* mice. All these observations indicate that *Lgr4* deficiency decreases the number of lipid absorptive cells.

2) The link between *Lgr4* and Notch signaling is still not clear. While more mechanistic studies in cell lines have shed some light, it is crucial to carry out functional experiments, including loss-of-function validations, in more relevant organoid or in-vivo systems to establish a clear connection between *Lgr4* deficiency and activated Notch signaling or clarify the gaps in knowledge regarding this in the text.

Reply: Per suggestion, we did more experiments to address the link between Lgr4 and Notch in organoids. The following data support that Lgr4 deficiency activates Notch signaling:

First, mRNA levels of Psen1 increased in organoids derived from Lgr4 deficient mice, which is consistent with the results observed in MODE-K and IEC6 cell lines (fig. S9e).

*Second, to confirm that the suppression of Wnt and activation of Notch result in the differentiation characteristics of *Lgr4^{iKO}* mice, we incubated organoids derived from WT mice with IWR-1 (Wnt inhibitor) and VPA (Notch activator) for one day, and found it could mimic the phenotype observed in *Lgr4^{iKO}* mice, at least to some extent (Fig. S9f and Fig. S9g).*

*Third, to further address the effect of Notch signaling, we incubated organoids derived from WT and *Lgr4* deficient mice with DAPT (Notch inhibitor). As shown in Fig. S9h-j, inhibition of Notch attenuated the effects of *Lgr4* deficiency on stem cell differentiation.*

*Lastly, we have tried several times to knock down Psen1 in organoids using siRNA or lentivirus following the instructions in literature, but could not get satisfactory knockout efficiency. Thus, we are not able to confirm the necessity of Psen1 for the phenotypes observed in *Lgr4* deficiency. This limitation has been addressed in lines 11-13, page 10.*

REVIEWERS' COMMENTS

Reviewer #1 (Remarks to the Author):

all questions have been addressed now and the paper should be accepted for publication

Reviewer #3 (Remarks to the Author):

The authors characterized Lgr4-deficient organoids, and technical challenges in creating knockdown/knockout of specific targets like Psen1 seem to exist. Overall, I have no further questions, as the authors have addressed most of my questions.